
# PARASO, a circum-Antarctic fully-coupled ice-sheet - ocean - sea-ice - atmosphere - land model involving f.ETISh1.7, NEMO3.6, LIM3.6, COSMO5.0 and CLM4.5

Charles Pelletier[1], Thierry Fichefet[1], Hugues Goosse[1], Konstanze Haubner[2], Samuel Helsen[3], Pierre-Vincent Huot[1], Christoph Kittel[4], François Klein[1], Sébastien Le clec'h[5], Nicole P. M. van Lipzig[3], Sylvain Marchi[3], François Massonnet[1], Pierre Mathiot[6,7], Ehsan Moravveji[3,8], Eduardo Moreno-Chamarro[9], Pablo Ortega[9], Frank Pattyn[2], Niels Souverijns[3,10], Guillian Van Achter[1], Sam Vanden Broucke[3], Alexander Vanhulle[5], Deborah Verfaillie[1], and Lars Zipf[2]

[1]Earth and Life Institute (ELI), UCLouvain, Louvain-la-Neuve, Belgium
[2]Laboratoire de Glaciologie, Université Libre de Bruxelles, Brussels, Belgium
[3]Department of Earth and Environmental Sciences, KU Leuven, Leuven, Belgium
[4]Laboratory of Climatology, Department of Geography, SPHERES, University of Liège, Liège, Belgium
[5]Earth System Science and Departement Geografie, Vrije Universiteit Brussel, Brussels, Belgium
[6]Met Office, Exeter, United Kingdom
[7]Université Grenoble Alpes/CNRS/IRD/G-INP, IGE, Grenoble, France
[8]ICTS, KU Leuven, Leuven, Belgium
[9]Barcelona Supercomputing Center (BSC), Barcelona, Spain
[10]Environmental Modelling Unit, Flemish Institute for Technological Research (VITO), Mol, Belgium

**Correspondence:** chrles.pelletier@protonmail.com

**Abstract.**

We introduce PARASO, a novel five-component fully-coupled regional climate model over an Antarctic circumpolar domain covering the full Southern Ocean. The state-of-the-art models used are f.ETISh1.7 (ice sheet), NEMO3.6 (ocean), LIM3.6 (sea ice), COSMO5.0 (atmosphere) and CLM4.5 (land), which are here run at an horizontal resolution close to $1/4°$. One key-

5 feature of this tool resides in a novel two-way coupling interface for representing ocean – ice-sheet interactions, through explicitly resolved ice-shelf cavities. The impact of atmospheric processes on the Antarctic ice sheet is also conveyed through computed COSMO-CLM – f.ETISh surface mass exchanges. In this technical paper, we briefly introduce each model's configuration and document the developments that were carried out in order to establish PARASO. The new offline-based NEMO – f.ETISh coupling interface is thoroughly described. Our developments also include a new surface tiling approach to combine

10 open-ocean and sea-ice covered cells within COSMO, which was required to make this model relevant in the context of coupled simulations in polar regions. We present results from a 2000 - 2001 coupled two-year experiment. PARASO is numerically stable and fully operational. The 2-year simulation conducted without fine tuning of the model reproduced the main expected features, although remaining systematic biases provide perspectives for further adjustment and development.



## 1 Introduction

The Antarctic climate is characterized by large natural fluctuations and complex interactions between the atmosphere, ocean, sea ice and ice sheet. One of the consequences of these interactions observed over the last decades lies in the Antarctic ice sheet (AIS) losing mass (Rignot et al., 2019), with the most spectacular changes located in West Antarctica (Shepherd et al., 2012). Atmospheric processes have been underlined as the main driver behind the large Antarctic climate variability (Jones et al., 2016; Raphael et al., 2016; Schlosser et al., 2018; Lenaerts et al., 2019). Since precipitation is the only source of mass gain for the AIS, moisture advection and atmospheric rivers have a direct impact on the evolution of the AIS (van Lipzig and van den Broeke, 2002; Gorodetskaya et al., 2014; Souverijns et al., 2018; Wille et al., 2021). However, the main source of AIS mass loss is the melting of the ice shelves (i.e., the floating part of the ice sheet), which are in direct contact with the Southern Ocean (Paolo et al., 2015; Jenkins et al., 2018; Shepherd et al., 2018). At the decadal timescale, atmospheric circulation patterns have an important indirect impact on ice-shelf melting through their direct influence on the oceanic circulation (Dutrieux et al., 2014; Jenkins et al., 2016; Holland et al., 2019). For example, oceanic processes such as the intrusion of relatively warm water over the Antarctic continental shelf, reaching the bottom of the ice shelves, have been suggested as an explanation for the recent increase in ice-shelf melt rates (Jacobs et al., 2011; Darelius et al., 2016; Rintoul et al., 2016). In turn, ice-shelf melting leads to freshwater injection at depth, thus impacting the circulation and heat transport in the Southern Ocean (Hellmer, 2004; Jourdain et al., 2017; Jeong et al., 2020). The impact on Antarctic sea-ice cover of this additional freshwater injection harbors complex phenomena which are still open to discussion. As a consequence, at the circumpolar scale, no definitive conclusion on the impact of this coupling mechanism has been drawn yet. While some preliminary studies suggested ice-shelf melting may have played a crucial role in the 1979 - 2016 Antarctic sea-ice extent (SIE) increase (Hellmer, 2004; Bintanja et al., 2013), other investigations have suggested that its impact is negligible (Pauling et al., 2016).

The examples listed above underline the importance of the interactions between the ice sheet, ocean (including sea ice) and atmosphere at the high latitudes of the Southern Hemisphere. Despite their varying typical response timescales, the respective evolutions of polar Earth subcomponents are entangled and interdependent (Turner et al., 2017; Fyke et al., 2018). Therefore, coupled models are the designated tool for quantitatively investigating them, since they allow representing the feedbacks between each component. As the success of coupled model intercomparison projects (e.g. CMIP6, Eyring et al., 2016) testifies, coupled ocean - atmosphere models have been a staple in climate modeling for a few decades. All state-of-the-art ocean models covering polar regions include a sea-ice submodel (see Bertino and Holland, 2017, for a review). More recent advances in ocean modeling (Dinniman et al., 2016; Asay-Davis et al., 2017) and the development of idealized test cases for simulating ice-shelf melting (Asay-Davis et al., 2016; Zhang et al., 2017; Jordan et al., 2018; Seroussi and Morlighem, 2018; Favier et al., 2019) have paved the way towards realistic ocean simulations with explicitly-resolved circulation within ice-shelf cavities on regional configurations (Donat-Magnin et al., 2017; Jourdain et al., 2017; Kusahara et al., 2017; Hazel and Stewart, 2020; Hausmann et al., 2020; Huot et al., 2021), but also on circumpolar domains (Naughten et al., 2018; Nakayama et al., 2020). Coupled ocean - ice-sheet configurations, with dynamical AIS geometry, have also been developed: using global climate models of intermediate complexity (Swingedouw et al., 2008; Goelzer et al., 2016; Ganopolski and Brovkin, 2017), but also state-of-the-



art ocean models over local domains (Seroussi et al., 2017; Timmermann and Goeller, 2017), and even at the global scale, for
pluri-centennial simulations relying on parameterized ice-shelf cavity circulation (Kreuzer et al., 2020). Model configurations
including ice-sheet coupling, with cavities explicitly resolved, are currently under development in a few global Earth system
models (Golaz et al., 2019; Sellar et al., 2019; Gierz et al., 2020), as well as in more idealized settings for performing sensitivity
studies (Richter et al., 2021).

In this paper, we introduce PARASO (PARAmour[1] Southern Ocean), a new five-component fully-coupled circumpolar
regional model covering the whole Antarctic continent and Southern Ocean. Using a regional (instead of global) configuration
allows us to focus on the specificities of this region at reasonable computational costs: about 2 simulated years per day with
approximately 250 cores, see App. B. Our tool relies on distinct models for each treated Earth system subcomponent: f.ETISh
v1.7 for the ice sheet (Pattyn, 2017; Sun et al., 2020); COSMO v5.0 for the atmospheric circulation, coupled to CLM v4.5 as a
land module (Rockel et al., 2008); NEMO v3.6 for the ocean (Madec et al., 2017), coupled to LIM v3.6 for the sea ice (Rousset
et al., 2015). A specificity of PARASO is that the ice-shelf melting is directly assessed from the explicitly-resolved oceanic
circulation within ice-shelf cavities. The ice-sheet model dynamically responds to these melt rates estimated from the cavities,
as well as to the surface mass balance (SMB) from the atmosphere and land models, by providing the ocean model an evolving
cavity geometry. The paper is organized as follows. In Sect. 2, each model component used in PARASO is briefly introduced.
The novel coupling interfaces specifically designed for our setup are more thoroughly discussed in Sect. 3. The configuration
and the external forcings are described in Sect. 4. Results from a two-year long fully-coupled simulation are commented in
Sect. 5. A broader discussion, along with concluding remarks, are provided in Sect. 6.

## 2 Model descriptions

### 2.1 Ice-sheet model: f.ETISh

The f.ETISh (fast Elementary Thermomechanical Ice Sheet) model v1.7 is a vertically integrated hybrid finite-difference ice
sheet/ice shelf model with vertically integrated thermomechanical coupling (see Pattyn, 2017, for a complete overview of
v1.0). While it is intrinsically a two-dimensional (plane) model, the transient englacial temperature field is calculated in a
three-dimensional fashion using shape functions for vertical velocities. Ice dynamics are solved through the combination of the
Shallow-Shelf Approximation (SSA) for basal sliding and the Shallow-Ice Approximation (SIA) for internal ice deformation
(Bueler and Brown, 2009). The marine boundary is represented by a grounding-line flux condition according to Schoof (2007),
coherent with power-law basal sliding. We employ a Weertman sliding law with exponent $m = 2$.

### 2.2 Ocean and sea-ice model: NEMO-LIM3.6

For simulating the ocean – sea-ice system, we use the Nucleus for European Modelling of the Ocean v.3.6 (NEMO3.6, Madec
et al., 2017), which includes OPA (Océan PArallélisé) for the open ocean coupled with the Louvain-la-Neuve sea-ice model

---

[1]PARAMOUR (Decadal Predictability and vAriability of polar climate: the Role of AtMosphere-Ocean-cryosphere mUltiscale inteRactions) - The name
of the project this tool has been developed for.





LIM3.6 (Vancoppenolle et al., 2012; Rousset et al., 2015). Hereinafter, this combination of ocean and sea-ice model is referred
to as NEMO.

### 2.2.1 Generalities on the NEMO3.6 open-ocean model

NEMO3.6 is a primitive-equation, free-surface, finite-difference, hydrostatic ocean model relying on a C-grid (Arakawa, 1966).
In our setting, the ocean dynamics are based on a split-explicit formulation. A $z^*$ vertical coordinate is used with varying cell
thickness over the whole column (Adcroft and Campin, 2004; Bruno et al., 2007). Vertical mixing is parameterized using
a turbulent kinetic energy scheme derived from Gaspar et al. (1990). Lateral diffusion of momentum is carried out with a
bi-Laplacian viscosity (Griffies and Hallberg, 2000), convection is parameterized as enhanced vertical diffusivity, and double-
diffusive mixing is included (Merryfield et al., 1999). A free-slip lateral boundary condition is applied at the coastlines and the
Beckmann and Döscher (1997) bottom boundary layer scheme is applied to both tracers and prognostic variables. The equation
of state is based on a 75-term polynomial approximation of TEOS-10 (IOC, 2010; Roquet et al., 2015).

### 2.2.2 Ice-shelf representation

In PARASO, Antarctic ice-shelf cavities are opened to the ocean circulation (Mathiot et al., 2017) with configuration-dependent
geometrical constraints defined in Sect. 4.1 and listed in Table 1. As in Barnier et al. (2006), partial (i.e., vertically cut) cells
are used to better represent the ice-shelf draft (depth of the immerged part of the ice sheet), the top cavity cell being defined as
the ice – ocean boundary layer (Losch, 2008). Heat and mass fluxes associated with melt are computed from the conservative
form of the three-equation formula from Hellmer and Olbers (1989); Holland and Jenkins (1999); Jenkins et al. (2010), relying
on velocity-dependent heat and salt exchange coefficients, here parameterized as in Dansereau et al. (2014):

$$\gamma_x = \sqrt{C_D^{\mathrm{isf}}} \| \mathbf{u}_h^{\mathrm{isf}} \| \Gamma_x \tag{1}$$

where $x \in \{T, S\}$ distinguishes heat and salt, $\Gamma_x$ is the default exchange coefficient, $C_D^{\mathrm{isf}}$ the ice – ocean drag coefficient, and
$\| \mathbf{u}_h^{\mathrm{isf}} \|$ the near-ice ocean current velocity. The ice-shelf melt parameterization also includes TEOS-10-compatible coefficients
for evaluating seawater freezing point temperature (Jourdain et al., 2017). All constants used in the NEMO ice-shelf module
are listed in Table 1.

### 2.2.3 Sea-ice model LIM3.6

The LIM3.6 sea-ice dynamics rely on an elastic-viscous-plastic rheology formulated on a C-grid (Bouillon et al., 2009, 2013).
Subgrid-scale sea-ice heterogeneity is rendered with a five-category ice-thickness distribution (Bitz et al., 2001), i.e., each
sea-ice field is five-fold, with distinct values over different ice-thickness ranges. The choice of five categories comes from a
tradeoff between computational constraints and physical realism in sea-ice mean state and variability (Massonnet et al., 2019;
Moreno-Chamarro et al., 2020). The sea-ice thermodynamics is based on an energy conserving scheme (Bitz and Lipscomb,
1999) and includes an explicit representation of the salt content of the sea ice (Vancoppenolle et al., 2009). The snow cover is
represented with one category defined on each ice thickness category (one snow category per ice category). The sea-ice surface





**Table 1.** NEMO ice-shelf-related physical constants. Here, "ice" refers to the ice sheet (not the sea ice). $\Gamma_T$ and $\Gamma_S$ are taken from Jourdain et al. (2017); $\rho_i$ is taken from f.ETISh, the ice-sheet model; all other melt thermodynamics parameters have been kept to their default NEMO values. All cavity geometry parameters have been set from a tradeoff between physical realism and numerical stability.

| | Name | Description | Value |
|---|---|---|---|
| Melt thermodynamics | $\rho_w$ | Ref. seawater density | $1026\ \mathrm{kg\,m^{-3}}$ |
| | $\rho_i$ | Ice density | $917\ \mathrm{kg\,m^{-3}}$ |
| | $\Lambda_f$ | Latent heat of fusion of ice | $3.34 \times 10^5\ \mathrm{J\,kg^{-1}}$ |
| | $c_p^w$ | Ocean specific heat | $3992\ \mathrm{J\,kg^{-1}\,K^{-1}}$ |
| | $c_p^i$ | Ice specific heat | $2000\ \mathrm{J\,kg^{-1}\,K^{-1}}$ |
| | $\mathcal{K}_i$ | Ice heat diffusivity | $1.54 \times 10^{-6}\ \mathrm{m^2\,s^{-1}}$ |
| | $C_d^{isf}$ | Top ice drag coefficient | $10^{-3}$ |
| | $\Gamma_T$ | Default heat exchange coef. | $2.21 \times 10^{-2}$ |
| | $\Gamma_S$ | Default salt exchange coef. | $6.19 \times 10^{-4}$ |
| Cavity geometry | $d_i^{min}$ | Min. ice-shelf draft | $10\ \mathrm{m}$ |
| | $h_w^{min}$ | Min. bathymetry | $20\ \mathrm{m}$ |
| | $h_i^{min}$ | Min. water column thickness in cavities | $50\ \mathrm{m}$ |
| | $h_i^{bl}$ | Ice–ocean boundary layer thickness | Top cell thickness |

albedo depends on ice surface temperature, ice thickness, snow depth and cloudiness (Grenfell and Perovich, 2004; Brandt et al., 2005). More details on LIM3.6 and the open-ocean – sea-ice coupling are given in Rousset et al. (2015) and Barthélemy et al. (2016), respectively.

## 2.3 Atmosphere and land model: COSMO-CLM²

To simulate the atmosphere, we use the version 5.0 of the COSMO-CLM Regional Climate Model. The COSMO model is
a Limited Area Model, developed by the COnsortium for Small-scale MOdeling (COSMO) and used both for Numerical Weather Predictions and in the context of Regional Climate Modeling. COSMO-CLM refers to the CLimate Mode (CLM) of the COSMO model that is used for the purpose of Regional Climate Modeling. The model is developed and maintained by the German Weather Service (DWD) and is now used by a broad community of scientists gathered within the CLM-Community (Rockel et al., 2008).
COSMO-CLM is a non-hydrostatic atmospheric regional climate model based on primitive thermo-hydrodynamical equations. The model equations are formulated on rotated geographical coordinates and a generalized terrain-following height coordinate is used (Doms and Baldauf, 2018). The COSMO-CLM model comprises several physical parameterization schemes representing a wide range of processes such as radiative transfer (Ritter and Geleyn, 1992), sub-gridscale turbulence (Raschendorfer, 2005) and cloud microphysics (Baldauf and Schulz, 2004).





COSMO-CLM is coupled to the Community Land Model version 4.5 (Oleson et al., 2013) using the OASIS 3-MCT coupler (Will et al., 2017). This coupling combination, which is hereinafter named CCLM$^2$, forms the land-atmosphere basis for PARASO, is described in Davin et al. (2011). More recently, an Antarctic CCLM$^2$ stand-alone configuration called AERO-CLOUD has been established. It features polar-specific recalibration of the atmosphere boundary layer scheme, the implementation of a new snow scheme, and the use of a two-moment microphysic scheme for precipitation (see Souverijns et al., 2019, for more detail). The PARASO CCLM$^2$ version has been adapted from AEROCLOUD, and their differences are detailed in App. C.

## 3 Coupling interfaces

In this section, we describe the three novel coupling interfaces developed for PARASO. Hereinafter, we call:

- *Coupling window*: a period between two coupling instances (intermodel exchange of information).

- *Restart leg*: a simulation "chunk", at the end of which the execution of the models is stopped and their content is written to disk, for the simulation to be pursued later as if it had not been interrupted. Restarts are typically used to reduce consecutive (timewise) CPU requirements and queuing time on supercomputers.

Generally speaking, NEMO and CCLM$^2$ run jointly with regular exchanges of information (online coupling, see Sect. 3.1). In between NEMO – CCLM$^2$ restart legs, f.ETISh runs by itself with input data files coming from NEMO and CCLM$^2$ outputs, and provides NEMO with updated information (offline coupling, see Sects. 3.2 and 3.3). Figure 1 offers a schematic view of the fully coupled system, with Fig. A1 giving more details on the timing strategy.

### 3.1 COSMO – NEMO coupling interface

#### 3.1.1 General description

In this section, we introduce the coupling interface between COSMO, the atmosphere model, and NEMO, the ocean – sea-ice model. We refer to "COSMO" (atmosphere model) instead of "CCLM$^2$" (atmosphere – land model), since this coupling interface does not involve land processes. COSMO and NEMO are coupled online (i.e., both model's executables are running simultaneously) through OASIS3-MCT_2.0 (Valcke et al., 2013; Valcke, 2013). While this coupling interface has been built from preexisting configurations involving these models (Van Pham et al., 2014; Will et al., 2017), our final setting includes several novel model developments which are described here. Figure 2 schematically represents this coupling interface.

Overall, NEMO sends COSMO surface values which are then used by COSMO to compute air-sea fluxes, eventually sent back to NEMO. The exchanged fields are detailed in Sect. 3.1.4. Spatial interpolations from one grid to the other are all performed bilinearly, except wind stresses which are interpolated bicubically to preserve curl. Despite being derived from the same initial f.ETISh geometry, the COSMO and NEMO continental masks may differ due to rounding effects at the coastline. In such cases, the nearest ocean neighbor is used to prevent COSMO from computing fluxes from land surface temperature with an ocean surface scheme, or NEMO from receiving fluxes that had been computed from the CLM land model.





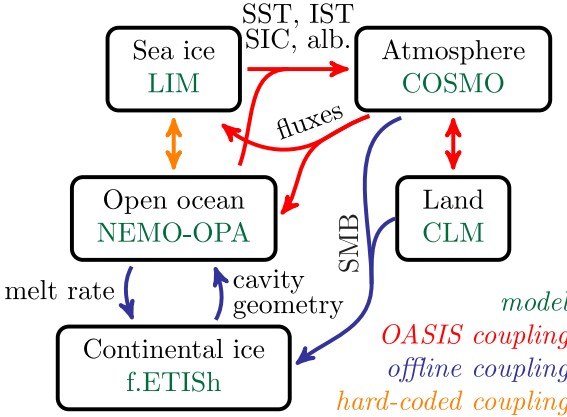

**Figure 1.** Fully coupled system submodel hierarchy. Each box represents an Earth system component and its corresponding model. The arrow colors distinguish the different coupling methods used, with legend given in the bottom right. "SST" stands for "sea surface temperature", "SIC" - "sea-ice concentration", "IST" - "ice surface temperature", "alb." - "albedo" and "SMB" - "surface mass balance". Above, "NEMO-OPA" represents the open-ocean part of NEMO (i.e., NEMO without LIM). Although being represented, the NEMO - COSMO exchanged fields are not detailed, as they have not been changed compared to existing settings.

The COSMO domain is smaller than NEMO's, hence the ocean surface boundary condition is coupled to COSMO if possible; else, outside of the COSMO domain, it is forced with atmospheric reanalyses (see Sect. 4.3 for more details). In order to avoid discontinuities and potential instabilities between both regimes, a linear transition over 20 NEMO grid points (the full grid being $\approx 600$ nodes) is applied southwards from the limit of the COSMO domain. For the sake of consistency, the same

reanalysis is used for forcing the COSMO lateral boundary condition and the extra-COSMO domain NEMO surface.

### 3.1.2 Timing strategy

The coupling frequency is set to the LIM time step $(5400\,\mathrm{s})$. Exchanged fields are time-averaged over the past coupling window by OASIS. The coupling is asynchronous: NEMO receives surface fluxes from COSMO with a one coupling window delay, which is less than the typical forcing frequency used for running either model in stand-alone. On the other hand, the surface

values seen by COSMO are real-time, albeit originating from NEMO which had been constrained by time-staggered fluxes. At the end of a restart leg, the content of the coupler is written to disk in order to guarantee smooth restartability. During the first coupling window of the first restart leg, NEMO receives fluxes obtained from monthly averages of a previous coupled run.

### 3.1.3 Tiling representation in COSMO

The main coupling-related development lies in the implementation of tiling in COSMO to represent heterogeneous ocean

surface cells, containing both open ocean and sea ice. Such methods have first been developed in land models (e.g. Koster and Suarez, 1992), then generalized to mixed surfaces (e.g. Best et al., 2004) and subsequently implemented in state-of-the-art





surface modules (e.g. Voldoire et al., 2017). The standard COSMO configuration requires one single set of surface values over each cell, which is then used to compute vertical diffusion. The general strategy is to have COSMO compute a set of two tile-specific surface fluxes, from two distinct sets of surface properties corresponding to the open ocean and the sea ice.

This is performed on all surface cells, regardless of the sea-ice concentration. The tile-specific fluxes are then sent to NEMO. The surface properties perceived by COSMO are degenerated from their tile-specific twofold values to a single set of values through pseudo-averaging. The pseudo-averaging consists in re-evaluating surface properties so that a given COSMO vertical column receives the concentration-wise average of the tile-specific fluxes it had computed. In that regard, the COSMO tiling implementation satisfies local air-sea energy conservation. Technical details on the pseudo-averaging operations are provided

in App. D.

### 3.1.4 Exchanged fields

Table 2 lists the exchanged fields between NEMO and COSMO. Sea-ice surface properties are averaged over thickness categories so that only one set of sea-ice surface properties is read by COSMO, and only one set of air – sea-ice fluxes is sent from COSMO to NEMO. As described in Rousset et al. (2015), surface fluxes injected into the sea ice are redistributed so that LIM

perceives distinct fluxes over each thickness category. With this redistribution, the net solar heat flux is the same as the one that would have arisen from a category-specific coupler and the net nonsolar heat flux is a first-order development (w.r.t. surface temperature) of the category-specific case. While the wind stress computations do not take into account surface velocities, their values above open ocean and sea ice are distinct as the surface roughness characterizations (and thus drag coefficients) differ. All heat fluxes are tile-specific and thus twofold. Over each tile, the solar heat flux relies on its tile-specific albedo, with the

sea-ice albedo taken as the average over ice thickness categories from a parameterization depending on ice and snow thicknesses (Grenfell and Perovich, 2004; Brandt et al., 2005). The tiled nonsolar heat fluxes contain net longwave (including direct downward atmosphere and upward surface contributions), sensible and latent contributions. The turbulent heat fluxes (sensible and latent heat) depend on surface properties and tile-specific turbulent heat transfer coefficients are evaluated from a turbulent kinetic energy (TKE) derived scheme (Raschendorfer, 2005). The latent heat fluxes are used for the diagnostic of sea-ice

sublimation and open-ocean evaporation. Sea-ice sublimation is permitted but solid condensation is not: in other words, over sea ice, downward latent heat can only be negative (i.e., it is set to zero if the surface scheme assumes it is positive). Rainfall is injected as freshwater at sea surface temperature; over sea ice, it is assumed to immediately runoff to the open ocean. Over sea ice, solid precipitation contributes to snow accumulation; over the open ocean, it is assumed to immediately melt, and the corresponding latent heat is removed from the local ocean surface cell. The computation of the surface temperature over sea ice

requires the nonsolar heat flux sensitivity (i.e., its derivative w.r.t. surface temperature) to ensure numerical stability, since the sea-ice surface temperature may significantly evolve within a coupling window (Iovino et al., 2013). The sensitivity estimate is local both time and space wise: it includes the temperature's influence on surface moisture, but assumes the heat transfer coefficient to be constant (which is a simplification). Fluxes over sea ice are systematically computed and sent over all ocean cells using the last known (resp., initial) sea-ice surface properties in case no sea ice is currently (resp., has ever been) present





**Table 2.** NEMO – COSMO list of exchanged variables.

| | Name | Unit |
|---|---|---|
| **NEMO → COSMO** | Sea surface temperature | K |
| | Sea-ice surface temperature | K |
| | Sea-ice albedo | - |
| | Sea-ice concentrations | - |
| **COSMO → NEMO** | Wind stress over ocean | $\mathrm{N\,m^{-2}}$ |
| | Wind stress over sea ice | $\mathrm{N\,m^{-2}}$ |
| | Net solar heat flux over ocean | $\mathrm{W\,m^{-2}}$ |
| | Net solar heat flux over sea ice | $\mathrm{W\,m^{-2}}$ |
| | Net nonsolar heat flux over ocean | $\mathrm{W\,m^{-2}}$ |
| | Net nonsolar heat flux over sea ice | $\mathrm{W\,m^{-2}}$ |
| | Nonsolar heat flux sensitivity over sea ice | $\mathrm{W\,m^{-2}\,K^{-1}}$ |
| | Snow/ice sublimation rate | $\mathrm{kg\,m^{-2}\,s^{-1}}$ |
| | Total evaporation/sublimation rate | $\mathrm{kg\,m^{-2}\,s^{-1}}$ |
| | Snowfall rate | $\mathrm{kg\,m^{-2}\,s^{-1}}$ |
| | Rainfall rate | $\mathrm{kg\,m^{-2}\,s^{-1}}$ |

in the category. This is carried out in case sea ice appears, through advection or thermodynamics, over one grid cell within a coupling window.

## 3.2   NEMO – f.ETISh coupling interface

### 3.2.1   General procedure

Figure 3 schematically represents the NEMO – f.ETISh coupling interface. Like most idealized and realistic settings (e.g.
Timmermann and Goeller, 2017; Favier et al., 2019), the coupling between NEMO and f.ETISh is asynchronous, sequential and restart-based. NEMO runs first for a coupling time window, sending f.ETISh monthly time series of ice-shelf melt fluxes (both melt and refreezing are permitted). Grounding line (including cavity opening and closing) and ice thickness are evolving freely with the minimum bound on ice-shelf thickness of 11m (very thin compared to typical values) to keep the ice-shelf extent constant, so that the PARASO land-sea mask does not evolve. Afterwards, NEMO awaits while f.ETISh runs for the
same coupling window, including the received melt rates as a boundary condition. After this, f.ETISh provides NEMO an updated ice-sheet geometry. NEMO then uses this new ice-shelf draft dataset starting from the beginning of the next coupling window. This process is iteratively repeated over each coupling window until the end of the simulation.



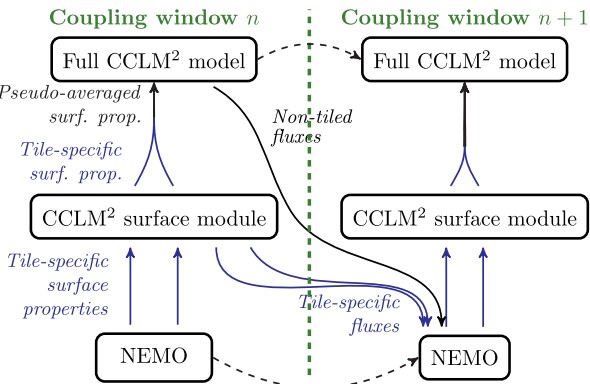

**Figure 2.** COSMO – NEMO coupling interface diagram for two coupling windows. Full blue (resp., black) arrows correspond to exchanges of tile-specific (resp., non-tiled) fields. The dashed green line delimits the transition from one coupling window to the next one, with the dashed black arrows representing each model's own dynamics. NEMO sends tile-specific surface properties to the COSMO surface module, which are used to compute tile-specific fluxes. These fluxes are then: (i) sent back to NEMO on the next coupling window (hence NEMO getting delayed fluxes); (ii) pseudo-averaged to be injected into the COSMO dynamics. In addition, non-tiled fluxes (e.g., precipitation) are also sent from COSMO to NEMO. For the sake of readability, the incoming (resp., outgoing) fluxes from coupling window $n-1$ (resp., to coupling window $n+2$) are not represented.

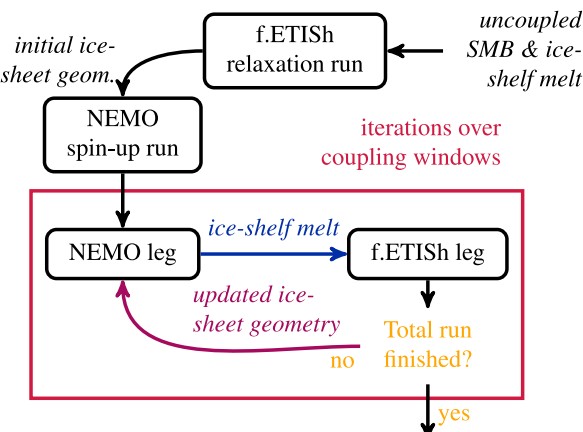

**Figure 3.** NEMO – f.ETISh coupling procedure. The red box represents the coupling loop (one iteration per NEMO – f.ETISh coupling window). Colored arrows represent the data exchanges between NEMO and f.ETISh. The NEMO spinup run is used as initial conditions for the first NEMO leg only.





### 3.2.2 Timing strategy

Our implementation is using a collaborative job submission script manager for NEMO called Coral. The coupling frequency
can be changed, as long as it is (all conditions must be met):

- an integer number of months;

- a multiple of the NEMO restart length;

- a divider of the total experiment length.

Regardless of the chosen coupling frequency, NEMO always provides monthly sub-shelf melt rates to f.ETISh to include
seasonality, and receives the simulated ice draft from f.ETISh at the end of the coupling step. For the results presented in Sect.
5, the NEMO-f.ETISh coupling frequency has been set to three months.

### 3.2.3 Exchanged fields

f.ETISh sends NEMO:

- two 2D boolean variables distinguishing (i) dry (grounded ice) and wet (ice shelves, open ocean) columns; (ii) an ice
mask (true for ice shelves and grounded ice);

- ice surface elevation;

- water column thickness.

These fields are then interpolated with a conservative method. The resulting columns are treated differently according to
the interpolated boolean variable values with a 50% threshold, as described in Fig. A2. With this procedure, grounding line
displacement is theoretically bounded by the NEMO grid southern limits, which, for PARASO, covers most of Antarctica (see
the NEMO grid imprint in Fig. 4).

NEMO sends f.ETISh monthly mean sub-shelf melt rates as freshwater mass fluxes $[\mathrm{kg\,m^{-2}\,s^{-1}}]$, which are converted to
ice equivalent melt rates $[\mathrm{m\,yr^{-1}}]$ by applying an ice density of $\rho_{\mathrm{ice}} = 917\,\mathrm{kg\,m^{-3}}$. The mask discrepancies between NEMO
and f.ETISh (e.g., arising from the NEMO-specific criteria for opening columns described in Sect. 4.1.2) are dealt with on a
cavity-by-cavity basis. If a f.ETISh cavity is at least partly covered by NEMO, the NEMO melt rates are interpolated bilinearly
(applied for about 84% of all ice-shelf grid cells in f.ETISh) with potential nearest-neighbor extrapolation to cover f.ETISh
cavity columns which correspond to closed columns in NEMO (applied for about 15% of all ice-shelf grid cells). Only a few
very small and dynamically insignificant ice shelves are not represented in NEMO, summing up to about 1% of all ice-shelf
grid cells in f.ETISh. Melt rates of cavity colums in NEMO which correspond to grounded ice or open ocean in f.ETISh
are disregarded (applicable for about 2.5% of the ice-shelf area in NEMO). The distribution between those procedures (84%
interpolation, 15% extrapolation and 1% not represented in NEMO) is dominated by the different grids of NEMO and f.ETISh
and stays almost constant even for longer runs (few decades, not shown here), the coupling frequency has only a very minor
influence on this phenomenon.





### 3.2.4 Procedure for opening and closing cavity cells in NEMO

After each coupling time step, NEMO uses the post-processed updated f.ETISh geometry as its new sub-shelf cavity geometry. The geometry constraints described in Table 1 are enforced at each coupling time step. The sea surface height, temperature and salinity of a newly opened ocean column are extrapolated from their neighboring cells (Favier et al., 2019). The extrapolation procedure is repeated 100 times, which yields a smoothening effect over potentially large new openings. The initial current velocities of new cells are set to zero. On all modified columns (thinned or thickened), an horizontal divergence correction

is applied in order to avoid spurious vertical velocities (Smith et al., 2021). Two caveats keep this post-coupling procedure from being conservative. First, energy and mass is brought into (resp., taken from) the ocean upon cell opening (resp. closing). Second, the divergence correction yields phantom current velocities into (resp., out of) grounded ice upon cell opening (resp., closing). Since PARASO is a regional configuration, with desired simulation times of a few decades at best, this lack of conservation has been deemed acceptable.

Geometry variations lead to spurious barotropic currents dissipating over the first few days following a mesh update. Practically, this implies that the coupling numerical stability is conditioned by the amplitude of the geometry variations through one update. However, we have observed that enforcing the NEMO cavity geometry constraints described in Table 1 keeps critical numerical instabilities from arising, even with yearly coupling windows (not shown here).

### 3.3  CCLM$^2$ – f.ETISh interface

The exchanges between CCLM$^2$ and f.ETISh are one-way (from CCLM$^2$ to f.ETISh), asynchronous, sequential and restart-based. CCLM$^2$ runs for one coupling time window, sending f.ETISh monthly time series of surface mass balance (SMB), which here boils down to solid precipitation minus surface sublimation. The SMB is converted from $[\mathrm{kg\,m^{-1}\,month^{-1}}]$ to ice equivalent thickness changes $[\frac{\mathrm{m}}{\mathrm{a}}]$ by using the reference ice density of $\rho_{\mathrm{ice}} = 917\mathrm{kg\,m^{-3}}$. Interpolation between the CCLM$^2$ and fETISh grids are performed bilinearly. For the sake of efficiency, the coupling window partitioning for the CCLM$^2$ –

f.ETISh interface is the same as the NEMO – f.ETISh one, and the workflow is managed with the same job submission tool (Coral).

The SMB received from CCLM$^2$ is included within the surface boundary condition of f.ETISh. However, variations in ice-sheet surface elevation are not sent back to CCLM$^2$, which is a limitation of PARASO, and the reason why this "coupling" procedure is referred to as "one-way". It should however be stressed out that at the time scales PARASO has been developed

for (decadal at the longest), variations in Antarctic surface topography are negligible.

## 4  The PARASO configuration

### 4.1  General description and geometry

In this subsection we detail each subcomponent of the PARASO configuration. Table 3 contains a brief, cross-model overview, and Fig. 4 shows and compares each model's configuration geometry.





**Table 3.** Summary of PARASO model setups. "res." stands for "resolution".

| Model | f.ETISh | NEMO | CCLM$^2$ |
|---|---|---|---|
| Earth component | Ice sheet | Ocean & sea ice | Atmosphere & soil |
| Domain boundary | Ice-shelf front (REMA[a]) | 30°S | Between 50°S and 40°S |
| Grid type | Stereographic proj. | Quasi-isotropic bipolar | Rotated lat-lon |
| Horizontal res. | 8 km | 0.25° | 0.22° |
| Vertical res. | - | 75 levels (1 m → 200 m) | 60 levels (10 m → 1000 m) |
| Time steps | 1/60 yr (525600 s) | 900 s (NEMO) & 5400 s (LIM) | 90 s (COSMO) & 3600 s (CLM) |
| Top forcing | - | ERA5 (outside of CCLM$^2$ domain) | ERA5 |
| Lateral forcing | - | ORAS5 | ERA5 |

[a]: Reference Elevation Model of Antarctica (Howat et al., 2019).

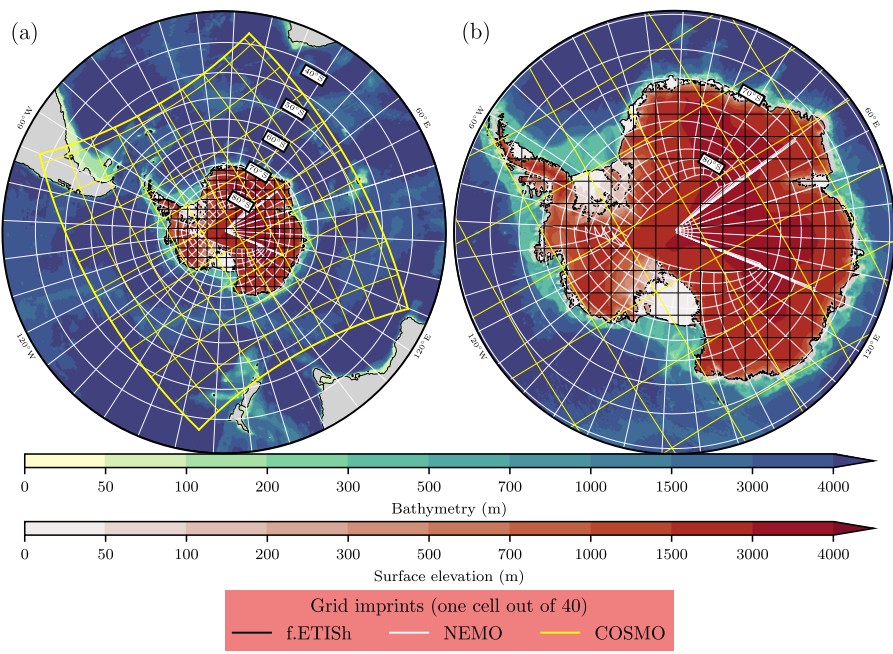

**Figure 4.** PARASO configuration geometry over (a) the full NEMO domain (cut at 30°S) and (b) Antarctica. The two colormaps indicate the bedrock bathymetry and the ice surface elevation (Morlighem et al., 2020) over Antarctica (note the nonlinear scale). The lines represent the f.ETISh, NEMO and COSMO grids (for the sake of readability, not all cells are drawn).

### 4.1.1 f.ETISh configuration

The f.ETISh model is run on a regular polar stereographic grid of $701 \times 701$ grid points centered around the South Pole, with a horizontal resolution of $8\,\mathrm{km}$. The f.ETISh time step is $1/60\,\mathrm{yr}$. The domain encompasses the continental shelf and shelf





break, which limits the maximum possible extension of the ice sheet. Bedrock topography, bathymetry, observed ice thickness and grounding-line position are taken from BedMachine Antarctica (Morlighem et al., 2020). This dataset, originally at a

500 m spatial resolution, has been resampled to the fETISh 8 km grid using a low-pass filter. Data was subsequently corrected to ensure grounded ice being actually grounded and floating ice being actually floating. The initial ice-sheet geometry for the PARASO setting is derived from the initialization and relaxation described in Sect. 4.2. For the PARASO configuration, iceberg calving has been omitted. The land ice extent is kept constant as the NEMO-COSMO interface has not been designed to deal with an evolving land-sea mask. Growing ice shelves are cut off at the calving front derived from BedMachine Antarctica

(Morlighem et al., 2020), but the equivalent iceberg melt flux is not injected into NEMO (an external iceberg melt dataset is used, see Sect. 4.3). Moreover, in accordance with the constraints listed in Table 1, shrinking ice shelves cannot become thinner than 11 m ($\approx$ 10 m draft, which is considerably smaller than typical draft values).

### 4.1.2   NEMO configuration

The NEMO PARASO set-up is derived from the GO7 configuration described in Storkey et al. (2018). The ocean grid is

ePERIANT025, which includes the southernmost ice-shelf cavities (Mathiot et al., 2017). With a single lateral ocean boundary at 30°S, NEMO has the widest spatial coverage of all PARASO subcomponents. The effective ocean model resolution decreases from 24 km at the 30°S boundary to 14 km at 60°S, eventually reaching 3.8 km in its southernmost cells (in the Ross cavity at approximately 86°S). The vertical discretization includes 75 levels with thickness increasing from 1 m at the surface to $\sim$ 200 m at depth. The NEMO and LIM time steps are 900 s and 5400 s, respectively. The Antarctic continental shelf bedrock

bathymetry is taken from BedMachine Antarctica (Morlighem et al., 2020) with a linear transition to ETOPO1 (Amante and Eakins, 2009) to cover the remaining PARASO domain (northwards from roughly 63°S). For numerical stability, a minimal ocean depth of 20 m is imposed. The Antarctic surface continental mask is constant timewise (calving is not permitted) and taken from the ice-sheet geometry obtained from the initial f.ETISh run described in Sect. 4.2. Ice-shelf cavities are opened to ocean circulation (Mathiot et al., 2017) with three geometrical constraints balancing physical realism and numerical stability

(see Table 1):

1. the minimal ice-shelf draft and water column height are set to 10 m and 50 m, respectively;

2. columns within ice-shelf cavities must have at least two vertical levels to allow overtuning circulation;

3. subglacial lakes (i.e., floating continental ice surrounded by grounded ice and separated from the ocean), ice-shelf crevasses and "chimneys" (vertical ocean segments surrounded by ice in all directions, and connected to the cavity)

are removed by NEMO (i.e. they are artificially filled with ice from their nearest-neighbor ice-shelf column).

Parameters enforcing these constraints are also listed in Table 1. At the depths of the ice-shelf cavities, the vertical resolution roughly ranges from 10 m to 150 m (but we remind that partial cells are used).





### 4.1.3 CCLM$^2$ configuration

In PARASO, the COSMO rotated latitude-longitude grid has a horizontal resolution of 0.22° (approximately 25 km at the

Antarctic coastline), covering the whole Antarctic ice sheet as well as a significant part of the Southern Ocean, with the northern boundary located between 50°S and 40°S. The terrain-following vertical discretization has 60 levels. The lowest model level is at 5 m height and cell thicknesses span from 10 m at the bottom to 1000 m at the top. The COSMO and CLM time steps are 90 s and 3600 s, respectively. The COSMO-CLM coupling is performed at every CLM time step (3600 s frequency).

### 4.2   Initialization

Prior to running the fully coupled system, a f.ETISh stand-alone initialization and relaxation run (see Sect. 4.2.1) is performed in order to generate a steady Antarctic ice-sheet state used as an initial geometry for all three subcomponents (see Fig. 4).

### 4.2.1   f.ETISh initialization

The f.ETISh model initialization is derived from an adapted iterative procedure based on Pollard and DeConto (2012) to fit the model as close as possible to present-day observed thickness (BedMachine Antarctica; Morlighem et al., 2020) and flow field by iteratively updating local basal friction coefficients (Pattyn, 2017). The method has been further optimized to enhance

convergence (Bernales et al., 2017). In addition to Pollard and DeConto (2012), we also implemented a regularization term that uses a 2D Gaussian smoothing kernel to filter out high-frequency noise in the basal sliding coefficients. For the initialization, f.ETISh is run with reduced complexity, using the shallow-ice approximation (SIA) for the evolution of the grounded ice sheet, thereby keeping ice shelves in the observed geometry. Optimized basal sliding coefficients and steady-state modeled geometry

for the Antarctic ice sheet were obtained after a forward-in-time integration of 60,000 yr with thermomechanical coupling. The initial temperature field (prior to that forward run) is derived from a steady-state temperature solution based on the present-day surface temperature, surface mass balance, and geothermal heat flux. The latter is based on the seismic-based dataset due to Shapiro and Ritzwoller (2004). For the relaxation, ice shelves are allowed to evolve and the hybrid shallow-shelf/shallow-ice approximation (HySSA; Bueler and Brown, 2009) is used to obtain the velocity field. The applied sub-shelf melt rates are

taken as the constant (timewise) 1984 - 1993 mean from a NEMO stand-alone run. The PARASO setting of f.ETISh (e.g. constant calving front, comp. Sect. 4) is run for 20 years to gain the relaxed geometry (see Fig. A8), representing the initial geometry for the PARASO configuration. The duration of 20 years is the result of a tradeoff between gaining quasi-steady-state ice-flow velocities and reducing the ice-shelf thinning during the relaxation. For the initialization and the relaxation, f.ETISh is forced by a constant surface temperature and SMB climatology provided by AEROCLOUD, a previous CCLM$^2$ stand-alone

experiment (Souverijns et al., 2019).

### 4.2.2   NEMO initialization

The ocean is initialized from a one-year spinup run performed with NEMO stand-alone, using the same forcing datasets as PARASO out of its coupling interfaces (i.e., ERA5 and ORAS5, see Sect. 4.3). The content in temperature and salinity for the


spinup is taken from the final 10-year climatology of a cavity-including 40-year NEMO stand-alone run, so that the spinup
is in equilibrium, in terms of both dynamics and thermodynamics. This allows PARASO to start from an upper ocean state
matching the CCLM² initialization, which is also based on ERA5.

### 4.2.3   CCLM² initialization

The land and ice masks used by CCLM² to determine the fraction of land vs. ocean and land ice vs. ice-free areas are calculated
on the CCLM² grid based on the geometry provided by f.ETISh, so that the geometry of the fully coupled setup is consistent
from one subcomponent to another.

COSMO 3D fields are initialized from the ERA5 atmospheric reanalysis (Hersbach et al., 2020). The CLM model starts from
initial conditions that are set internally in the code Oleson et al. (2013), implying that the land component is not in equilibrium
with the atmosphere at the initial time. The visible and near-infrared albedos for glacier ice are respectively set to 0.80 and
0.55. Glacier temperatures are initialized at 250 K. The overlying snow pack is modeled with up to five layers, depending
on the total snow depth. At the initialization, five layers are used to represent a snow water equivalent (SWE) of 1 m, which
corresponds to a 4 m thick snow layer for a bulk snow density of 250 kg m⁻³. The snow liquid water and ice contents for
layer $i = 1$ to 5 are set to $w_{liq,i} = 0$ kg m⁻² and $w_{ice,i} = \Delta z_i \rho_{sno}$ kg m⁻², respectively. Although a 1 m SWE snow pack is too
small to fully represent the firn layer, which can extend to more than one hundred meters in Antarctica (van den Broeke et al.,
2009), Souverijns et al. (2019) showed that it leads to a decent SMB representation in CCLM², as long as the adaptations from
van Kampenhout et al. (2017) are included to the CLM snow module (except for the snow pack depth). For future PARASO-
related work, keeping a maximum snow depth of 1 m is also convenient to limit the spin-up phase of the snow pack to a
decade. A higher maximum snow depth would extent the spin-up phase, but also the risk of simulating permanent snow cover
(difficult to correct for thicker pack) in places which are normally seasonally capped by snow (van Kampenhout et al., 2017).
Multiple surface datasets are also required to initialize CLM. They are listed in Table 2.5 of Oleson et al. (2013) and comes
from a variety of sources. Depending on the domain (location, resolution), datasets can be downloaded and regridded on a
case by case basis using the CESM tools (the procedure is described in Chapter 1 of  Vertenstein et al., 2013). Note that one
specificity of PARASO lies in the modifications of the land mask and mean elevation, carried out to be consistent with the
f.ETISh relaxation run (see Fig. 5). In particular, the ice-shelf surface elevation is represented, while the original CESM data
simply assumes all ice shelves to be exactly at sea level.

## 4.3   External forcing

### 4.3.1   f.ETISh forcings

Sub-shelf melt rates and SMB are provided as coupled fields by NEMO and CCLM².

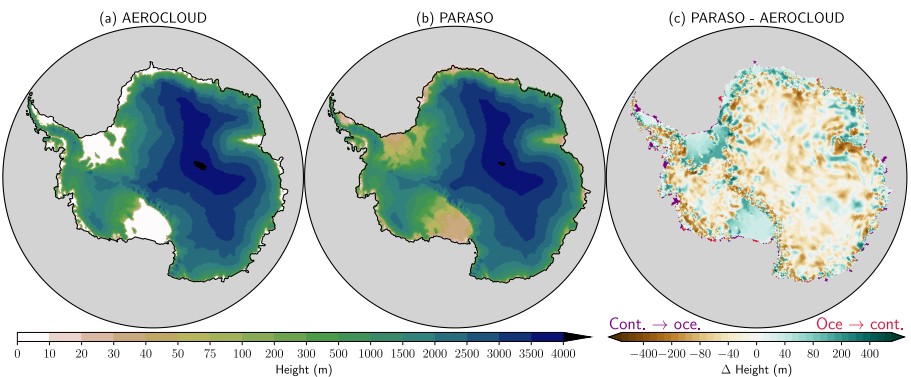

**Figure 5.** Surface elevation for (a) a previous CCLM$^2$ stand-alone model configuration (AEROCLOUD), (b) PARASO and (c) their absolute difference (PARASO - AEROCLOUD). Note that all color scales are nonlinear. On (c), nodes that change nature, from AEROCLOUD land to PARASO ocean (resp. AEROCLOUD ocean to PARASO land), are displayed with purple (resp. red) shadings.

### 4.3.2 NEMO forcings

Over the ocean surface located outside of the CCLM$^2$ spatial coverage, the NEMO surface is forced with fluxes computed by
the CORE bulk formula (Large and Yeager, 2004), using the ERA5 reanalysis as atmosphere input (Hersbach et al., 2020). At its 30°S lateral open ocean boundary, NEMO is constrained with the NEMO-based ORAS5 ocean reanalysis (Zuo et al., 2019) at monthly frequency. The lateral boundary condition is enforced with a Flather radiation scheme for 2D dynamics (SSH and barotropic velocities) and a flow relaxation scheme for the baroclinic velocities, temperature and salinity. A constant $86.4 \times 10^{-3}$ W m$^{-2}$ geothermal flux is imposed at the bottom of the ocean (Stein and Stein, 1992). River runoff is adapted from
Dai and Trenberth (2002) and prescribed as a climatological data set (Bourdallé-Badie and Treguier, 2006). While icebergs are not dynamically simulated, their freshwater melt input is injected as additional runoff from an interannual dataset obtained from a previous iceberg-including NEMO simulation covering 1979 to 2015 (Marsh et al., 2015; Merino et al., 2016; Jourdain et al., 2019). Buoyant plume mixing resulting from runoff is parameterized with enhanced mixing in the shallowest 10 m, over the river deltas and on a stripe between 20 km and 200 km from the Antarctic coast (where most of the iceberg melting occurs).
Enhanced mixing is excluded from the direct vicinity of the coast to avoid interferences with ice shelf meltwater (N. Jourdain, personal communication, 2019). No salinity restoring is applied.

### 4.3.3 CCLM$^2$ forcings

At its lateral boundaries, COSMO is relaxed to ERA5 using a one-way interactive nesting based on Davies (1976). This consists in defining a relaxation zone where the internal COSMO-CLM solution is nudged against the external ERA5 solution. Within
this zone, the variables of the driving model are gradually combined with their corresponding variables in COSMO-CLM by adding a relaxation forcing term to the tendency equations that govern their evolution. The attenuation function which specifies the lateral boundary relaxation inside the boundary zone has a tangent hyperbolic form (Källberg, 1977). The width




of the relaxation layer is set to 220 km, which corresponds to approximately 10 times the typical cell size. More information about the one-way nesting can be found in the COSMO documentation (Doms and Baldauf, 2018). The boundary forcings
(as well as the initial conditions) for the COSMO model have been prepared at discrete time intervals using the interpolation program INT2LM (Schättler and Blahak, 2013). The interval between two consecutive sets of boundary data (frequency of the lateral forcing) is 3 h. Within this time interval, boundary values are interpolated linearly in time. 3D atmospheric variables for COSMO are wind speeds (zonal and meridional), air temperature, pressure deviation from a reference pressure (1000 hPa at sea level), specific water vapour content, specific cloud water content and specific cloud ice content. No spectral nudging
has been applied to the upper atmosphere model levels to preserve the CCLM$^2$ atmosphere dynamics. A sponge layer with Rayleigh damping in the upper levels of the model domain is activated in order to avoid artificial reflections of gravity waves. A cosine damping profile with maximum damping at the top and zero damping at the base of the sponge layer, that is 11000 m, is assumed.

## 5  Results

In this section, diagnostics from a 2-year (2000 - 2001) PARASO experiment are presented and evaluated. Unless explicitly specified otherwise, the results shown are taken from the second simulated year (2001) in order to limit the influence of the initialization. A summary of all experiments designed for this study and used for diagnostics is provided in Table 4.

**Table 4.** List of experiments specifically designed for this study and used in diagnostics. Besides the presence (or lack thereof) of coupling interfaces, all experiments share the same design and cover the 2000 - 2001 period. In Sect. 5.3, PARASO is also compared to AEROCLOUD (a previous CCLM$^2$ stand-alone experiment, see Souverijns et al., 2019), which is not listed here since it has not been developed for comparison with PARASO, and does not share the same experimental design. The three PARCRYO experiments only differ in their forcing. PARCRYO$^{CTRL}$ is forced with the constant SMB used during the initialization (provided by AEROCLOUD; see Sect. 4.2.1) and no sub-shelf melt is applied. PARCRYO$^{SMB}$ also does not apply sub-shelf melt and is forced with monthly SMB from PARATMO. PARCRYO is forced with monthly SMB from PARATMO and monthly sub-shelf melt rates from PAROCE.

| Name | CCLM$^2$ | NEMO | f.ETISh |
|---|---|---|---|
| PARATMO$^a$ | Y | N | N |
| PAROCE | N | Y | N |
| PARCRYO$^{CTRL}$ | N | N | Y |
| PARCRYO$^{SMB}$ | N | N | Y |
| PARCRYO | N | N | Y |
| PARASO | Y | Y | Y |

$^a$: same experimental design as PARASO except for the surface flux tiling which has not been implemented in CCLM$^2$ stand-alone.





The PARASO biases are more generally discussed and put into perspective in Sect. 6. Computational aspects related to PARASO are briefly covered in App. B. PARASO-specific tunings used for obtaining a realistic sea-ice seasonal cycle are
described in App. E.

## 5.1 Ice sheet and ice shelves

Figure 6 displays the sub-shelf melt rates for PAROCE and PARASO in different regions. The regional melt rate patterns coincide in both experiments (Figs. 6(b)-(e)), but the total melt of PARASO is significantly increased for all East Antarctic regions (Weddell Sea, Indian Ocean, West Pacific; Figs. 6(a)-(c) and 6(f)) due to the warmer and less saline Eastern Antarctic
surface water in PARASO (see Figs. 6(g), 9 and 10). However, the West-Antarctic melt rates have the same magnitude in PARASO and PAROCE (Figs. 6(a) and 6(d)-(e)). Compared to observed ice-shelf melt rates from Rignot et al. (2013) and Adusumilli et al. (2020), both experiments underestimate sub-shelf melt. While the simulated melt rates for the Ross and Weddell Seas match observations, the melt for the West Pacific sector, Bellingshausen and Amundsen Sea in particular, is considerably lower (Fig. 6(a)). The under-estimated melt rates stem likely from increased ice shelf geometry changes during
the initialization (see Sect. 4.2 and Fig. A8). A similar NEMO stand-alone run (not shown here) with an observation-based ice-sheet geometry (from BedMachine Antarctica, Morlighem et al., 2020) provides significant larger melt rates for those regions, in better agreement with Rignot et al. (2013) and Adusumilli et al. (2020). These results support that the geometry is a main driver of the observed melt biases. The importance of cavity geometry on simulated melt rates has already been emphasized by previous studies (Schodlok et al., 2012; Rosier et al., 2018; Goldberg et al., 2019; Brisbourne et al., 2020; Goldberg et al.,
2020; Wei et al., 2020).

Figure 7 compares PARASO and the PARCRYO experiments, corresponding uncoupled stand-alone f.ETISh simulations for different forcings (see Table 4). Overall, PARCRYO and PARASO display a similar behavior across most coastal regions, and in particular the Antarctic Peninsula, with a moderate grounded ice thickness increase up to a few meters (Figs. 7 (b)-(c)). That thickening also occurs for PARCRYO$^{SMB}$ (Fig. 7 (d)) and correlates well with the differences between the forcing SMB
data sets (Fig. 7 (g)). PARACRYO$^{SMB}$ and PARACRYO apply the SMB from PARATMO, while PARACRYO$^{CTRL}$ is forced by the SMB based on AEROCLOUD that has been used for the f.ETISh initialization (see Sect. 4.2.1). The direct comparison of PARCRYO and PARASO (see Fig. 7(f)) reveals a slightly larger thickening of the grounded ice in coastal areas for PARASO. This is in accordance with the increased SMB for those regions provided by CCLM$^2$ in PARASO (Fig. 7(i)). Figure 7(j) also emphasizes the correlation between the SMB forcing and the thickness evolution of the grounded ice. PARCRYO and
PARCRYO$^{SMB}$, which are forced by the identical SMB data set (from PARATMO), show a very similar evolution of the grounded ice (Fig. 7(e)), while the change of the volume above floatation (VAF) for PARASO clearly follows the enhanced SMB. A more detailed discussion of the differences between AEROCLOUD, PARATMO and PARASO can be found in Sect. 5.3. Other ice thickness changes occur at the grounding line (Figs. 7(a)-(c)), as the relatively coarse resolution and numerical uncertainties of the grounding line flux induce small oscillations between neighboring grid cells in the grounding line position.
Furthermore, the experiments show a thickening of many narrow ice streams (e.g. Pine Island Glacier, Byrd Glacier, Rutford



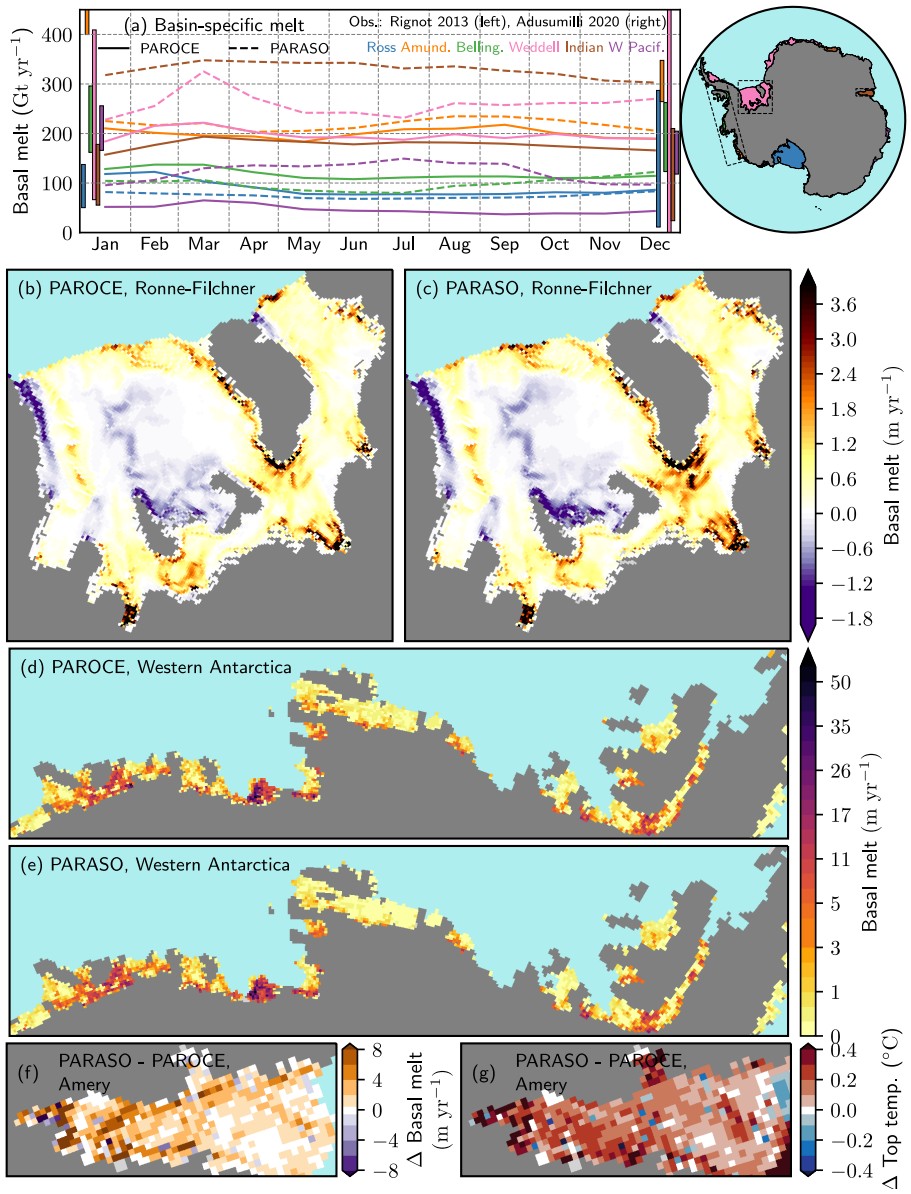

**Figure 6.** (a): Total ice-shelf melt rate assessed from PAROCE (dashed) and PARASO (solid), over distinct basins (color, see legend on figure and indicative map), all sub-figures for the year 2001 (second simulated year). (a) features observations of ice-shelf melt rates and uncertainties over the same basins: left-hand side from Rignot et al. (2013), right-hand side from Adusumilli et al. (2020). (b) - (c): zoom-in on the Ronne-Filchner cavity for (b) PAROCE and (c) PARASO (negative values indicate ice-shelf refreezing). (d) - (e): zoom-in on the Western Antarctic sector for (d) PAROCE and (e) PARASO (note the nonlinear colormap scale). (f) - (g): zoom-in on the Amery ice shelf for (f) melt rate and (g) top conservative temperature PARASO - PAROCE differences.



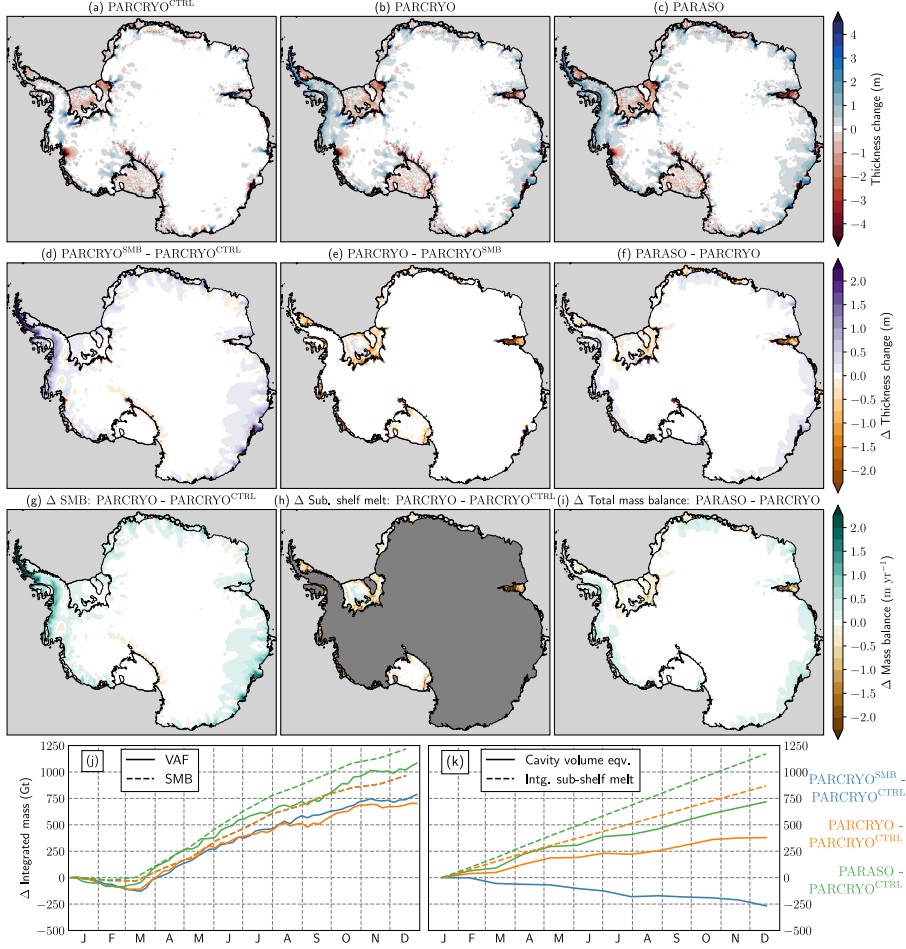

**Figure 7.** (a) - (c): Ice thickness change during the year 2001 for (a) PARCRYO[CTRL], (b) PARCRYO and (c) PARASO. (d) - (f): Difference of the ice thickness change between (d) PARCRYO[SMB] and PARCRYO[CTRL], (e) PARCRYO and PARCRYO[SMB] and (f) PARASO and PAR-CRYO. (g): Difference between the mean SMB applied for PARCRYO (SMB based on PARATMO) and for PARCRYO[CTRL] (SMB from initialization run based on AEROCLOUD). (h): Mean sub-shelf melt rates (from PAROCE) applied for PARCRYO – no melt rates applied in PARCRYO[CTRL]. Negative values indicate melt, positive values indicate refreezing. (i) Difference of the total mass balance between the coupled PARASO and the stand-alone PARCRYO experiment. (j) Change of the volume above floatation (VAF) and the integrated SMB and (k) change of the cavity volume and the integrated sub-shelf melt. (j) - (k): Plotted are PARCRYO[SMB] (red), PARCRYO (blue) and PARASO (black) as anomaly compared to PARCRYO[CTRL]. For simplicity, all variables are given in gigatons. All sub-figures represent simulations result of the year 2001 (second simulated year).

Ice Stream), and only Thwaites Glacier shows significant grounded ice mass loss (Figs. 7(a)-(c)). The suspicious thickening is due to both the limited resolution and the initialization (see Sect. 4.2.1) and causes a certain model drift.





The forcing with sub-shelf melt, which has only been applied for PARCRYO (sub-shelf melt from PAROCE) and PARASO, has a very limited effect on the grounded ice on the short evaluation period of one year, but clearly affects the ice shelves
(Fig. 7(e)). The ice shelves show a moderate thinning of up to a few meters almost everywhere, except in the Bellingshausen Sea, where substantial ice-shelf thickening occurs (Figs. 7 (b)-(c)), as the increase of the SMB compared to PARCRYO$^{\text{CTRL}}$ exceeds the applied sub-shelf melt (Figs. 7(g) and (h)), while for the other regions the sub-shelf melt is the dominating forcing for the ice shelves. Figure 7(k) illustrates the influence of the different SMB forcing on the ice shelves. The enhanced SMB of PARCRYO$^{\text{SMB}}$ compared to PARCRYO$^{\text{CTRL}}$ leads to a decrease of the cavity volume, hence ice shelf thickening. Figure 7(k)
also exhibits enhanced ice-shelf thinning (stronger cavity volume increase) of PARASO compared to PARCRYO, following the increased sub-shelf melt of PARASO. The increased ice shelf thinning of PARASO mainly occurs across the East Antarctica ice sheet (Fig. 7(f)) and is in accordance with the increased sub-shelf melt of PARASO, which also mainly affects the East Antarctic ice shelves (Figs. 6(a) and 7(i)).

In terms of ice-ocean coupling, the ice sheet behaves very similarly for both coupled and uncoupled experiments (PARASO
and PARCRYO, respectively), and their differences are explained by discrepancies in ice-sheet forcing (SMB from CCLM$^2$, sub-shelf melt rates from NEMO). No biases or additional noise due to the coupling itself have been found for the ice sheet model. It should be kept in mind, however, that both simulations cover only two years, which is a short period for the relatively slow (e.g., in comparison with the ocean and atmosphere) ice-sheet system.

## 5.2 Ocean and sea ice

In this section, we compare PARASO results to observations as well as to PAROCE, a NEMO stand-alone experiment forced by ERA5, with the same experimental design (except the coupling, see Table 4).

Figure 8 displays sea-ice observations (sea-ice index from the National Snow & Ice Data Center, Fetterer et al., 2017) and diagnostics from PAROCE and PARASO. Generally speaking, the PARASO Antarctic sea-ice extent seasonal cycle shares similar features compared to PAROCE (see Fig. 8(a)), with larger biases. The chronology of the first-year cycle (i.e., dates of
maximum and minimum) is well simulated, still with both configurations retaining too little sea ice in the summer (minimum extent at $\approx 10^6$ km$^2$ instead of $3 \times 10^6$km$^2$), which is a persistent bias for most coupled models (Turner et al., 2013; Mahlstein et al., 2013; Roach et al., 2018, 2020). PAROCE and PARASO also suffer from another well-known bias in forced NEMO simulations, displaying too large maximal Antarctic sea-ice extent and too rapid subsequent melting (Vancoppenolle et al., 2009; Rousset et al., 2015; Barthélemy et al., 2018). The second-year PARASO seasonal cycle is considerably degraded compared
with the first year, with a more significant low bias in maximum sea-ice extent ($16 \times 10^6$ km$^2$ instead of $18 \times 10^6$ km$^2$). In contrast with the sea-ice extent, the PARASO maximum sea-ice volume is generally larger than PAROCE's (see Fig. 8(b)). This may be linked to the relatively stronger PARASO winds and their coastwards orientations, making sea ice more compact but less extensive (see Fig. A7). Regarding the sea-ice growth cycle (Figs. 8(c)-(e)), the PAROCE and PARASO biases are similar, with the PARASO growth being delayed by about 15 days compared to PAROCE, which then translates into smaller maximum
sea-ice extent. On the other hand, in many regions, the PARASO sea-ice decay cycle (Figs. 8(f)-(h)) occurs much earlier than PAROCE. The clearest examples are the Indian and Western Pacific sectors, whose decay cycles start from a too small cover,





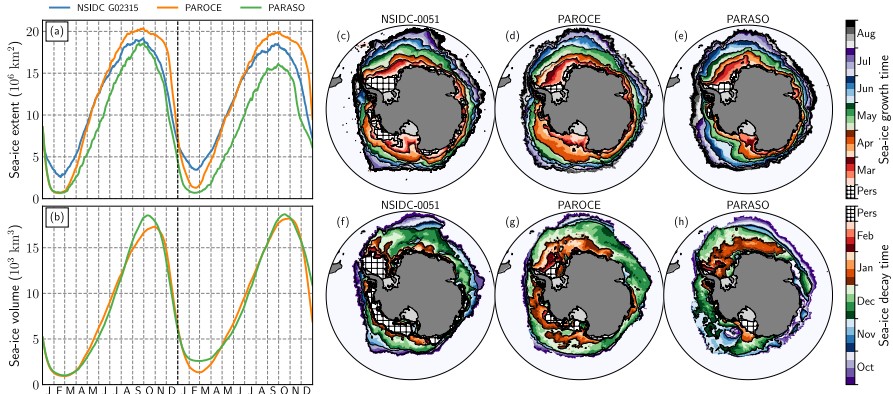

**Figure 8.** (a) Daily sea-ice extent from observations (NSIDC-G02315), PAROCE and PARASO, over 2000-2001. (b) Same for sea-ice volume (no observations available). (c)-(e): Sea-ice growth progression over the 2000 freezing season from (c) observations (NSIDC-0051), (d) PAROCE and (e) PARASO. The shadings indicate the earliest time of the year at which sea ice appears after the yearly minimum (in February 2000). (f)-(h): Same for the sea-ice decay progression over the 2000-2001 melting season. The shadings indicate the earliest time of the year at which sea ice disappears after the yearly maximum (early September 2000). On both colorbars, "Pers" indicates persisting sea ice throughout each full freezing or melting process. Sea-ice presence is defined by a 15% concentration threshold.

and where the formation of coastal polynyas linked to strong katabatic winds (see Fig. A7) contributes to leaving only a thin sea-ice strip from mid-December on. In Western Antarctica, virtually no sea ice is present in PARASO from mid-December on, whereas in observations it can persist year-round (and in PAROCE, melt one month later). In the Ross and Bellingshausen

Seas, most of the sea-ice melt is occurring more than a month earlier in PARASO compared with PAROCE, which is in better agreement with observations. In comparison with the aforementioned regions, the PARASO sea-ice pack in the Weddell sector melts later, but still leaves smaller persisting sea-ice cover compared to PAROCE. The only exception to this rapid PARASO sea-ice decay lies in front of the Ross ice shelf, where strong winds blowing towards the coast lead to the formation of a large and thick persisting sea-ice pack (see Fig. A7).

Figures 9 and 10 show in-situ temperature and practical salinity diagnostics from the regridded World Ocean Atlas 2018 (Boyer et al., 2018), as well as those from PAROCE and PARASO. The PARASO temperature and salinity biases are generally similar to PAROCE's, even close to the air-sea interface (Figs. 9(a)-(b) and 10(a)-(b)). This can be explained by the NEMO initialization, which starts from a biased mean state corresponding to this specific configuration's equilibrium. It should however be noted that in PARASO, the Eastern Antarctic subsurface is perceivably warmer and less saline than in PAROCE, matching

the increased ice-shelf melt rates observed in Sect. 5.1. This is also related to the very rapid sea-ice decay in that area: from early spring, the ocean surface can undergo warming and the enhanced sea-ice melting leads to additional early freshwater release. An examination of the vertical profiles drawn in Figs. 9(c)-(n) and 10 confirms that the main origin of PARASO biases are related to the initialization, since they are similar to PAROCE. In general, deep waters are too cold ($\Delta T \approx -1°C$) and fresh ($\Delta P_s \approx 0.25 \mathrm{psu}$) in both PAROCE and PARASO. One possible cause for this bias is the presence of ice-shelf melt water

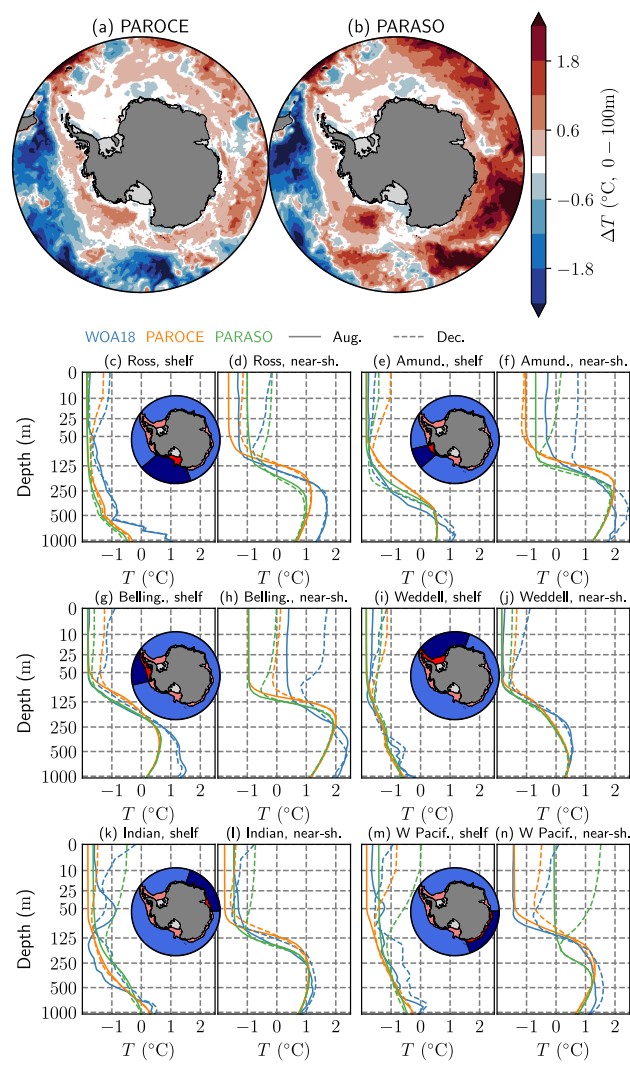

**Figure 9.** (a) - (b): In-situ temperature biases for (a) PAROCE and (b) PARASO, with respect to WOA18, averaged on the top 100 m of the ocean and the full seasonal cycle. (c) - (n): in-situ temperature vertical profiles for WOA18, PAROCE and PARASO, in August (full) and December (dashed), over six circumpolar basins drawn on indicative maps. Note the nonlinear $z$-axis. On the indicative maps, bright red shadings indicate the continental shelf (defined by the 1200m isobath, see (c),(e),(g),(i),(k),(m)); dark blue shadings indicate the "near-shelf" (see (d), (f), (h), (j), (l), (n)), defined as the area outside of the continental shelf and south of $60°$S. For WOA18, the data is taken from a 1995 - 2004 climatology; for PAROCE and PARASO, from simulation outputs for the year 2001 (second simulated year).

injection at depth, which is a novel feature for both configurations. While these general biases are present, the vertical structure of the water column is represented well within PARASO, and interseasonal changes are of the same magnitude as in WOA18.



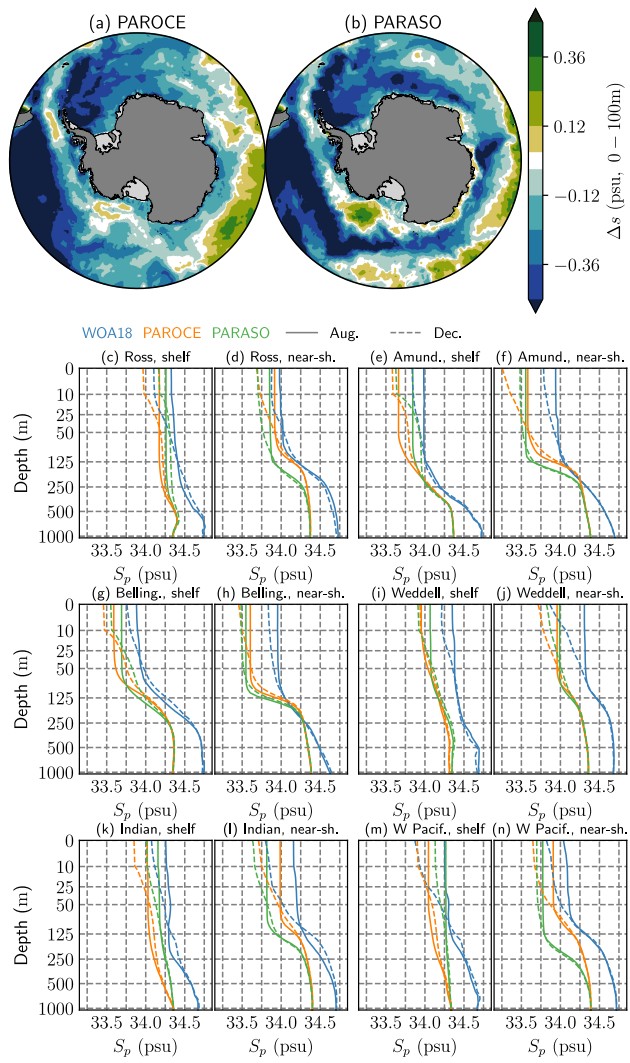

**Figure 10.** (a) - (b): Practical salinity biases for (a) PAROCE and (b) PARASO, with respect to WOA18, averaged on the top 100 m of the ocean and the full seasonal cycle. (c) - (n): practical salinity vertical profiles for WOA18, PAROCE and PARASO, in August (full) and December (dashed), over six circumpolar basins (same as Fig. 9). Note the nonlinear $z$-axis (following the NEMO vertical discretization). For WOA18, the data is taken from a 1995 - 2004 climatology; for PAROCE and PARASO, from simulation outputs for the year 2001 (second simulated year).

In 2001, the Antarctic Circumpolar Current (ACC) transport through the Drake Passage is $47 \pm 4$ Sv ($1$ Sv $= 10^6$ m$^3$ s$^{-1}$) in PAROCE and $43 \pm 4$ Sv in PARASO (see also Fig. A6). This is severely underestimated compared with observations (e.g., $173.3 \pm 10.7$ Sv in Donohue et al., 2016) or other simulations (e.g., 117 Sv in Mathiot et al., 2011). A source of Drake ACC weakening in PARASO is the presence of spurious westwards currents along the Antarctic continental shelf (not shown here),



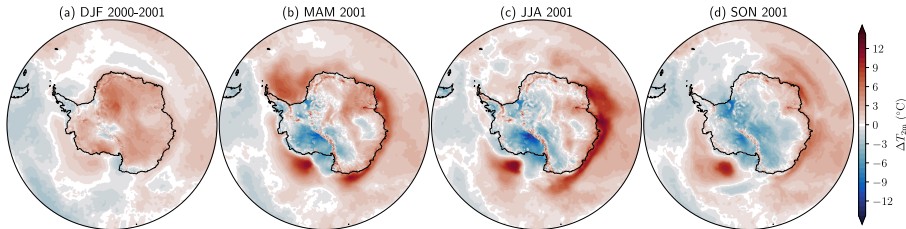

**Figure 11.** (PARASO - ERA5) differences in seasonally averaged 2 m air temperature for the second simulated year.

counterbalancing the eastwards transport and thus leading to this degraded net transport. While both PAROCE and PARASO Drake transport values display large biases, which will be a focus of developments in the forthcoming of the model, the fact that the values are not significantly different from each other suggests that the coupling itself does not bear a significant impact on ACC transport.

## 5.3 Atmosphere

Figure 11 shows seasonally-averaged 2 m air temperature differences between PARASO and ERA5. Large differences (up to $15°C$ in absolute value) are found over the ice shelves and the ocean close to Antarctica. The differences are much smaller in summer (Fig. 11(a)) compared to the winter season (Fig. 11(c)), when more sea ice is present and the atmosphere over the ice shelves is very stable. Over the Ross and Filchner-Ronne ice shelves, the 2 m air temperature is systematically lower in PARASO than in ERA5. This systematic difference is too large to be solely attributed to differences in elevation between PARASO and ERA5 over the ice shelves, but this effect might contribute to the temperature signal visible over the Berkner and Roosevelt islands. Due to the scarcity of in-situ observations over the ice shelves, a fraction of the difference between PARASO and ERA5 may be due to uncertainties in ERA5 (Gossart et al., 2019). Moreover, differences of comparable magnitude, but of opposite sign, are observed when comparing PARASO with the Japanese 55-year Reanalysis (JRA-55, see Fig. A3). PARASO was also compared to AEROCLOUD, a pre-existing CCLM$^2$ stand-alone experiment that was thoroughly evaluated using in-situ observational data (not shown, Souverijns et al., 2019). It was found that both experiments behave similarly on the Filchner-Ronne ice shelf. The comparison to AEROCLOUD over the Ross ice shelf, in contrast, provides similar results to ERA5. The boundary layer over the ice shelves is very stable and the large differences between ERA5, JRA-55, PARASO and AEROCLOUD are probably related to deficiencies in the representation of turbulent fluxes in the boundary layer, which is known to be a considerable challenge in polar regions. Due to the lack of data over the ice shelves, it is difficult to definitely attribute this deficiency to one of the products mentioned above.

Over the ocean, the (PARASO - ERA5) differences are largest in the East Antarctic sector, where the 2 m air temperature is systematically higher in PARASO than ERA5. This is consistent with the increased ice-shelf melt rates (see Sect. 5.1) and near-surface ocean warming (see Sect. 5.2) previously highlighted in this sector. In addition to ERA5 and AEROCLOUD, PARASO was also compared to PARATMO (see Table 4), a CCLM$^2$ stand-alone run with the same tuning paramaters as for PARASO, except for the surface flux tiling which has not been implemented for CCLM$^2$ stand-alone experiments (instead,





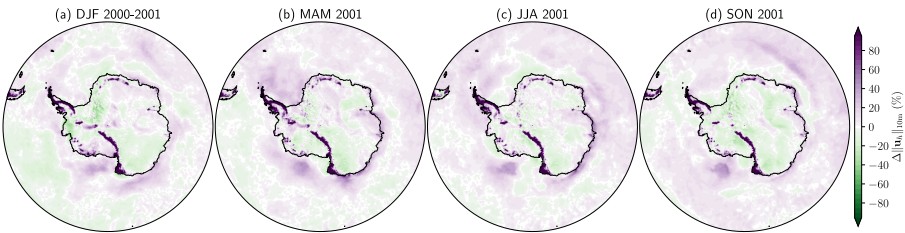

**Figure 12.** Relative differences in seasonally averaged 10 m horizontal wind speed norm between ERA5 and PARASO (i.e., (PARASO-ERA5)/ERA5).

sea ice is assumed to fully cover a grid cell where the surface temperature is lower than $-1.8\,°C$). The large East Antarctic warm difference is less prevalent in PARATMO (see Fig. A4), thus suggesting an origin related to coupling. Moreover, in PARASO, large positive temperature anomalies are simulated over a restricted area in the Ross Sea. They are associated with
the development of an open-ocean polynya, where the excessive vertical mixing induces the release of a substantial amount of energy. Some deep convection and open-ocean polynya formation have occurred during the past decades in the Southern Ocean, but many atmosphere - ocean - sea-ice coupled climate models that participated in the latest CMIP6 simulate them too frequently and at locations where they have not been observed (Heuzé, 2021), such as in the Ross Sea in PARASO.

Figure 12 displays the relative differences in seasonally-averaged 10 m horizontal wind speed between PARASO and
ERA5. Major differences are located in the Antarctic Peninsula and the Transantarctic Mountains. There, the magnitude of the PARASO winds can be more than twice the value in ERA5, with little interseasonality differences. More generally, the regions of highest wind speed differences match the regions where the elevation differs the most. The effect of subgrid scale orography is accounted for through an effective roughness length (Lott and Miller, 1997), which is likely lower in PARASO compared to ERA5. Over the continent, outside of mountainous regions, the surface wind speed is generally lower in PARASO
than ERA5. Gossart et al. (2019) showed that ERA5 underestimates the 10 m wind speed in coastal areas and the interior of Antarctica. Similar conclusions apply to PARASO.

The differences in PARASO and ERA5 SMB, calculated as total precipitation minus surface sublimation, are displayed on Fig. 13. Precipitation is the dominant source of the Antarctic ice-sheet SMB. Accordingly, the differences in SMB between PARASO and ERA5 primarily result from differences in the simulated precipitation. As surface melt runoff (only significant
for areas below 1000 m elevation) is not included and snowdrift processes are not represented, the estimated SMB may depart from observations. However, the scarcity of in-situ observations prevents us from accurately estimating the AIS SMB and we chose to compare our SMB estimate to ERA5, whose quality is discussed in Gossart et al. (2019). Moreover, in Mottram et al. (2021), SMB estimates from different Antarctic regional climate models (including CCLM[2]) are compared, and the observed intermodel spread suggests considerable uncertainties even in uncoupled, atmosphere-only simulations.

Compared to ERA5, PARASO provides higher SMB values over East Antarctica and the Weddell Peninsula (Figure 13). For East Antarctica, the largest differences are located in the coastal areas, where the SMB in PARASO can be more than twice the SMB in ERA5. Those differences may partly result from a too low SMB estimate in ERA5, particularly below 500m



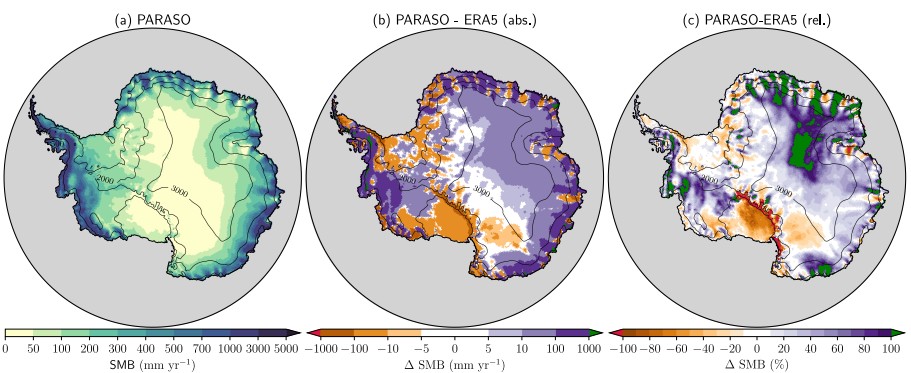

**Figure 13.** Surface mass balance reconstruction for 2001, the second simulated year. (a) PARASO; (b) PARASO absolute difference w.r.t. ERA5; (c) PARASO relative difference w.r.t. ERA5. Note the nonlinear color scales. Contours denote elevation at 500 m, 2000 m and 3000 m above mean sea level.

elevation (Gossart et al., 2019). Souverijns et al. (2019) found, when investigating the spatial pattern of the simulated SMB in AEROCLOUD with the reconstruction based on ice cores and ERA-Interim of Medley and Thomas (2018), a significant underestimation of the SMB for most of the coastal sites including the Antarctic Peninsula. A comparison of PARASO to AEROCLOUD reveals higher SMB values in those regions, suggesting a better agreement with observations. Irrespective of the comparison product used (AEROCLOUD or ERA5), PARASO simulates lower SMB values over the Ross ice shelf. This is primarily attributed to fewer precipitation in PARASO over this region (not shown). We also analysed the SMB provided by PARATMO (see Fig. A5), for which similar results to PARASO are found. This argues that the model behavior can at least be partly related to the representation of atmospheric dynamics, and not solely to the coupling.

## 6 Discussion and conclusions

In this technical paper, we have introduced PARASO, a new five-component coupled configuration for simulating the Earth system in the high latitudes of the Southern Hemisphere. Aside from the novel coupling interfaces, establishing this new tool required substantial model developments in the COSMO atmosphere model, in order to adapt its surface scheme to requirements from NEMO (mostly related to flux tiling), the ocean – sea-ice model it is coupled to in PARASO. To our knowledge, PARASO is the first publicized circumpolar Antarctic ocean - atmosphere coupled regional configuration that includes an ice-sheet model, with ice-shelf cavities explicitly resolved. The ocean – ice-sheet offline coupling interface has also been thoroughly described in this paper. However, our results suggest that at the short timescales investigated for this technical paper, the practical impact of this particular coupling interface is minor, and that the main features of PARASO would be reproduced with a similar NEMO - CCLM$^2$ coupled configuration (i.e., excluding coupling with f.ETISh).

In addition to the functioning of the coupling interfaces, the major PARASO achievement lies in its numerical stability. Though, the biases observed in PARASO are a significant drawback, which prevents PARASO from being considerably more





skillful than some global coupled configurations. Global climate models suffer from similar biases in the high latitudes of the Southern Hemisphere (Wang et al., 2014; Schneider and Reusch, 2016) and the PARASO biases are of comparable magnitude

to the differences between distinct reanalyses (see Figs. 11 and A3). Overall, PARASO is a novel tool and further calibration could reduce the biases. For this first evaluation of our tool, the objective was to check whether the biases were affected by the coupling interfaces themselves (which they are not), rather than to correct the more general issue related to each model's biases. Extra tuning for limiting biases introduced by our choice of model combination and data input changes is however of high priority. While recognizing that clearly distinguishing each PARASO component's contribution to the biases from those

purely due to their coupling interfaces is far from straightforward, we consider the former to be beyond the scope of this study.

Besides the coupling itself, the PARASO biases may be imputable to changes in each model's configuration that led to apply each component in conditions different from the ones in which they were developed and calibrated. Yet, our choice has been to retain these changes, since they are relevant. For example, taking the Antarctic surface topography from f.ETISh as initial $CCLM^2$ geometry instead of the standard data set made the PARASO setup more consistent intermodel-wise. How-

ever, $CCLM^2$ parameters had been tuned to this standard topographical data set, and this might have impacted the results. In other words, we have preferred fundamental consistency over the precise agreement with current observations. Incorporating our novelties while limiting biases also identifies clear paths for model improvements. Below we provide three potential bias sources and perspectives related to their corrections or limitations. We also refer to App. E for more details on tuning experiments which have led to PARASO, as the "out-of-the-box" coupling between PARASO components led to even stronger

biases.

First, in the vicinity of their common interface, the ocean and atmosphere are highly sensitive to their boundary conditions (Miller et al., 1992; Large et al., 1997; Torres et al., 2019a, b). For PARASO, the interface boundary condition has been altered on both sides for enforcing COSMO – NEMO intermodel compatibility. Besides being assessed from a dynamical model instead of an external dataset (such as a reanalysis), ocean - atmosphere fluxes are computed differently in PARASO compared

to either NEMO or COSMO stand-alone. On the NEMO side, the ocean receives turbulent fluxes from the COSMO surface scheme instead of an ocean-calibrated bulk formula (e.g., CORE Large and Yeager, 2004). Performing NEMO stand-alone experiments with a COSMO-derived bulk formulation is a considerable challenge, as the COSMO surface scheme requires several near-surface atmosphere properties (e.g. TKE at bottom levels or laminar transfer coefficients) as input. On the COSMO side, the atmosphere receives tiled fluxes which harbor subgrid-scale heterogeneity in surface properties. Since the COSMO surface

scheme has been tuned for untiled fluxes (it uses instead a binary approach), the new tiling method, which is a requirement for coupling with NEMO, may lead to undesired systematic biases. Performing COSMO stand-alone experiments with tiled fluxes has not been pursued because this would have required further adapting the COSMO sources and forcing datasets. In a recent study, Heinemann et al. (2021) present a nonlinear tiling approach for COSMO (similar to that of PARASO) as an enhancement of a non-conserving method (Gutjahr et al., 2016), which also relies on sea-ice thickness (not considered in PARASO). This

may be of use for future studies intending at investigating the sole impact of nonbinary tiling. Alternative approaches are possible to locally reduce the biases in the air-sea fluxes, in particular by nudging them to reanalysis data, as done for instance in Ho-Hagemann et al. (2020) in a recent study coupling NEMO and COSMO. This has the advantage of providing model results





closer to observations, but at the cost of perturbing the atmosphere – ocean feedbacks and adding artificial energy sources and sinks at the surface, thus limiting the applications of the model outside of present-day conditions.

Second, coupled PARASO experiments suffer from excessive latent heat (and thus evaporation) over sea ice, as illustrated in Fig. E1(e). Since coupled and forced snowfall rates are roughly similar (not shown), this evaporation bias is the main driver beneath the observed decrease in snow thickness over sea ice (see Fig. E1(c)). This leads to a positive feedback further inhibiting the presence of snow over sea ice, and thus the radiative balance at the sea-ice surface: the excess of evaporation degrades the snow cover, the surface albedo is reduced, more solar radiation is absorbed by the sea-ice – snow system, hence,

even more snow melts at the surface. While some specific tuning (see App. E) helped reducing this bias, its positive feedback has not been fully controlled, and this aspect represents one of the main challenges for future PARASO developments.

Third, another potentially large source of biases is the updated Antarctic continent geometry used in PARASO. As described in Sect. 4.2, the initial Antarctic topography is derived from a f.ETISh relaxation run, which displays clear differences from observations, related to this model's specific biases and to the forcing data sets it relied on (SMB from COSMO, ice-shelf

melt rates from NEMO). Out of inter-model consistency, all PARASO initial geometries have been directly derived from this f.ETISh relaxation run. In the atmosphere, the signature of the new topography can be seen on the differences between AEROCLOUD and PARASO (see Fig. 5 and Sect. 5.3). In the ocean, the near-Antarctic bathymetry is crucial to the warm water intrusion into the cavities, and thus to the ice-shelf melt rates (see Goldberg et al., 2019, for an overview). Results from 30 yr NEMO – f.ETISh coupled experiments (without CCLM$^2$, see Fig. A9) have suggested that at least on these timescales,

the initial cavity geometry plays a much bigger role on ice-shelf melt rates than the ice sheet – ocean coupling. This is to be expected, since the melt rates induced by the coupling on decadal time scales - and therefore the simulated cavity changes - are smaller than the ice geometry changes during the multi-millenial f.ETISh initialization (see Sect. 4.2).

The three sources of biases listed above have been mostly identified from comparison with the different stand-alone versions of each used model. However, other bias sources enhanced by the coupled nature of PARASO may also be present within this

tool. As previously mentioned, the objective of this paper is to document our tool, assess its performance as-is, and share it with the community. Taking advantage of the numerical stability of PARASO and the expertise gained in its development process, additional tuning and calibration experiments aiming for further bias reduction are currently at the designing stage. Finally, it may be worth noting that in comparison with the short experiments of Sect. 5, the sea-ice extent biases remain stable or are even reduced with time in a longer fully-coupled PARASO experiment (see Fig. A10). This suggests that the ocean surface

warming described in Sect. 5 remains circumscribed to relatively small amplitudes, hence that PARASO does not keep on drifting at the longer term, and that its performance remains reasonably satisfactory at least in terms of Antarctic sea ice.

*Code and data availability.* Complete sources for PARASO are available free of charge to members of the CLM-Community: https://wiki. coast.hzg.de/clmcom/, as tarball number XXX. In addition to this tarball, all PARASO sources, at the exception of their COSMO-CLM parts, are freely available from https://doi.org/10.5281/zenodo.5337510. All PARASO input data for running a three-month long PARASO



experiences are available from https://doi.org/10.5281/zenodo.5223540 and https://doi.org/10.5281/zenodo.5342468. Scripts and data for generating the figures contained herein are available at https://doi.org/10.5281/zenodo.5337520.

The COSMO-CLM model is free of charge for all research applications; however, access is license-restricted (see http://www.cosmo-model.org/content/consortium/licencing.htm, last access: August 24th 2021). To download, the user needs to become a member of the CLM-Community or the respective institute needs to hold an institutional license. The documentation of the COSMO model is maintained by the COSMO-Consortium at https://www.dwd.de/EN/ourservices/cosmo_documentation/cosmo_documentation.html (last access: August 24th 2021).

NEMO, LIM and XIOS (a NEMO-compatible I/O library) are developed by the NEMO consortium, and distributed under the Ce-CILL license. The NEMO-LIM version used in PARASO has been built from the standard 3.6 version, with the ice-shelf following two modifications: (i) an undocumented lateral sea-ice melt scheme (J. Raulier, UCLouvain); (ii) the ice-shelf coupling module from the revision 11248 of the dev_isf_remapping_UKESM_GO6package_r9314 NEMO development branch, available from https://forge.ipsl.jussieu.fr/nemo/browser/branches/UKMO (last access: August 24th 2021). Complete NEMO documentation is available from the NEMO consortium website: www.nemo-ocean.eu.

f.ETISh (Fast Elementary Thermomechanical Ice Sheet model of intermediate complexity v1.7) has been developed by F. Pattyn and co-workers (Pattyn, 2017). This program is free software: you can redistribute it and/or modify it under the terms of the GNU General Public License as published by the Free Software Foundation, either version 3 of the License, or (at your option) any later version.

The OASIS-MCT coupler is developed by the CERFACS (Toulouse, France) and the CNRS (Paris, France). It is distributed under the Lesser GNU General Public License (LGPL). OASIS-MCT_2.0 can be downloaded from the its official website: https://portal.enes.org/oasis (last access: August 24th 2021).

The Coral job submission tool is developed by the CISM in the UCLouvain (Louvain-la-Neuve, Belgium) and is distributed under the Creative Commons CC0 1.0 Universal license.

The ocean – ice sheet coupling interface also relies on: the Climate Data Operators (CDO, Schulzweida, 2019) version 1.9.5, developed at the Max-Planck-Institute for Meteorology (see https://code.mpimet.mpg.de/projects/cdo/, last access August 24th 2021), and released under the terms of the GNU General Public License v2 (GPL); the netCDF Operators (NCO, see Zender, 2021) version 4.7.6, developed at the University of California, Irvine, and distributed under the GNU GPL v2 license (see http://nco.sourceforge.net, last access August 24th 2021).

All input data used for running PARASO, besides the lateral COSMO forcings, are available at www.zenodo.org/XXXXX. Due to their prohibitive size, the lateral COSMO forcings are not publicly available, but they are available upon request to the corresponding author.

Scripts and data for reproducing the figures contained in this manuscript are available from zenodo.org/XXXXXX.



# Appendix A: Additional figures

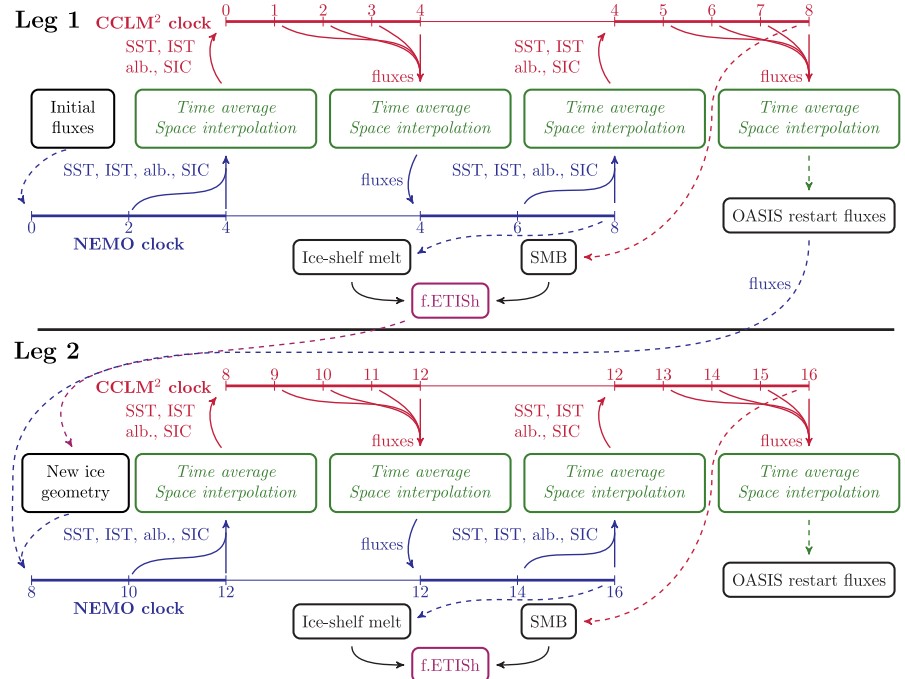

**Figure A1.** Fully coupled system timing scheme. Here, two restart legs are represented, separated by the thick horizontal black line. The blue and red timelines represent the NEMO and CCLM physical clocks, respectively. They are deliberately staggered to account for NEMO receiving delayed fluxes. For readability, it is assumed that the CCLM time step is 1, NEMO's is 2, the NEMO-CCLM coupling frequency is 4 and the ice sheet model coupling frequency is 8. Inbetween legs, the NEMO-CCLM system is stopped. Black boxes represent data written on the disk. Green boxes represent OASIS operations. Full arrows represent exchanges of data between models and OASIS. Dashed lines represent reading or writing to disk. "SST" stands for sea surface temperature; "IST", ice surface temperature; "alb.", albedo; "SIC", sea ice concentration; "SMB", surface mass balance.

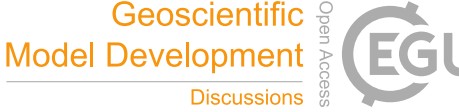

Perform **conserv_interp**(is_ice,is_wet,ice_thickness,
            ice_elevation,bedrock_bathy,init_draft)
**for** $(i,j) \in$ NEMO columns **do**
  **if** is_ice$(i,j) \geq 0.5$ **then**
    **if** is_wet$(i,j) \geq 0.5$ **then**          *Ice, wet $\rightarrow$ cavity*
      draft$(i,j) =$ ice_thickness$(i,j) -$ ice_elevation$(i,j)$
    **else**                 *Ice, dry $\rightarrow$ grounded ice*
      draft$(i,j) =$ bedrock_bathy$(i,j)$
  **else**                          *No ice*
    **if** is_wet$(i,j) \geq 0.5$ **then**     *No ice, wet $\rightarrow$ open ocean*
      draft$(i,j) = 0$
    **else**                  *No ice, dry $\rightarrow$ bedrock continent*
      draft$(i,j) = 0$
      bedrock_bathy$(i,j) = 0$
              *f.ETISh keeps the front from moving but*
              *the post-processing can lead to such cases.*
  **if** draft$(i,j) > 0$ **and** init_draft$(i,j) = 0$ **then**
    draft$(i,j) = 0$          *Force retreat advancing ice-shelf front*
  **else if** draft$(i,j) = 0$ **and** init_draft$(i,j) > 0$ **then**
    draft$(i,j) = 10$ m         *Force fill retreating ice-shelf front*

**Figure A2.** Pseudocode for the ice sheet geometry post-processing. "**conserv_interp**" is a conservative interpolation from f.ETISh's grid to NEMO's. On the f.ETISH grid, "is_ice" is 0 for open ocean column, else (cavity or grounded ice) it is 1; "is_wet" is 0 for grounded ice, else (cavity or ocean) it is 1. "draft" is the ice-shelf draft; like the bedrock bathymetry, it is positive downwards and set to zero at the ocean surface. "ice_elevation" is the ice surface elevation compared to floating point level, defined positive upwards. "init_draft", the initial NEMO ice-shelf draft, is used to ensure that the procedure has not led to ice shelf front displacement.

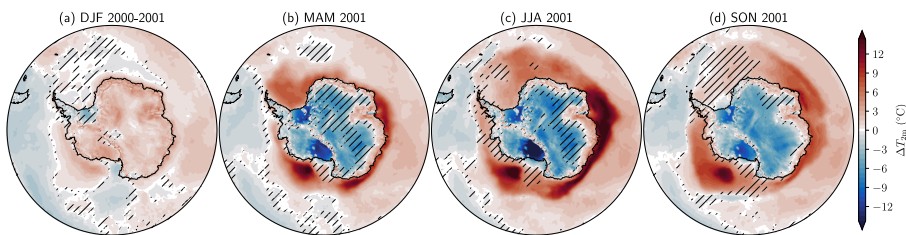

**Figure A3.** (PARASO - JRA-55) (Japanese 55-year Reanalysis, Kobayashi et al., 2015) differences in seasonally averaged 2m air temperature for the second simulated year. The used colorbar is the same as in Fig. 11. Hatching denotes areas where the (PARASO - JRA-55) and (PARASO - ERA5) (see Fig. 11) differences are of contrary signs.



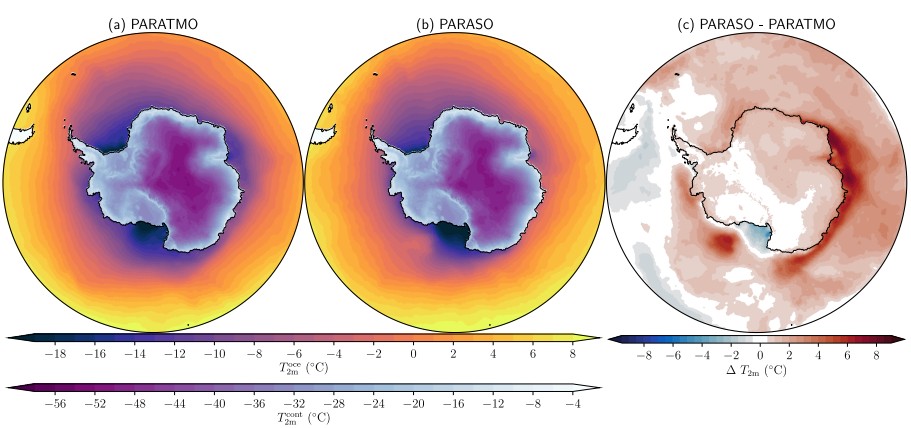

**Figure A4.** 2001 average of 2m air temperature for (a) PARATMO, (b) PARASO and (c) (PARASO - PARATMO) difference. Note that on (a)-(b), distinct color scales are used for the ocean ("oce") and continent ("cont").

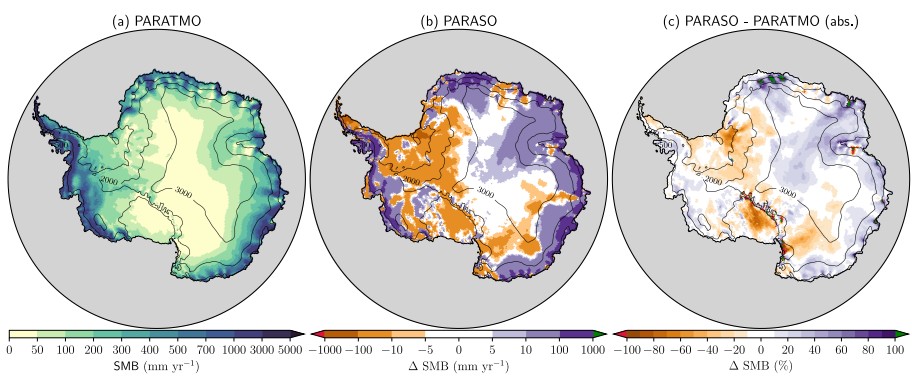

**Figure A5.** 2001 average of SMB for (a) PARATMO, (b) (PARASO - PARATMO) absolute difference and (c) (PARASO - PARATMO) relative difference. Note the nonlinear color scales. Contours denote elevation at 500 m, 2000 m and 3000 m above mean sea level.



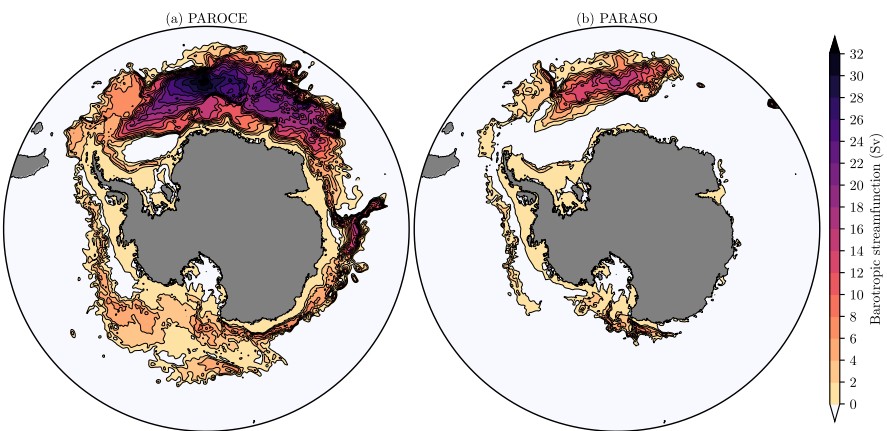

**Figure A6.** Barotropic streamfunction for (a) PAROCE and (b) PARASO, averaged over 2001 (second simulated year), in Sv (1 Sv = $10^6$ m$^3$ s$^{-1}$). The streamfunction has been set to zero over Antarctica. Positive values indicate clockwise circulation. Negative values are not represented.

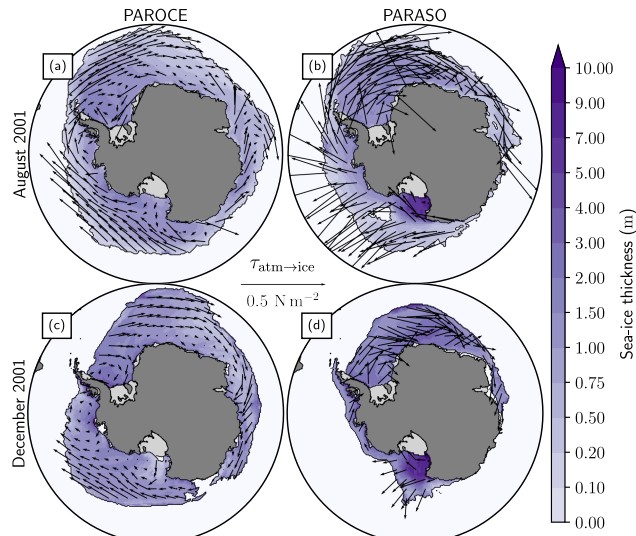

**Figure A7.** Sea-ice thickness (colormap) and air – sea-ice wind stress (arrows) in (a),(b) August and (c),(d) December of 2001 (second simulated year), for (a),(c) PAROCE and (b),(d) PARASO. Sea-ice thicknesses are only drawn on areas with sea-ice concentrations exceeding 15%. While the wind stress scale is linear, note that the sea-ice thickness's is not.

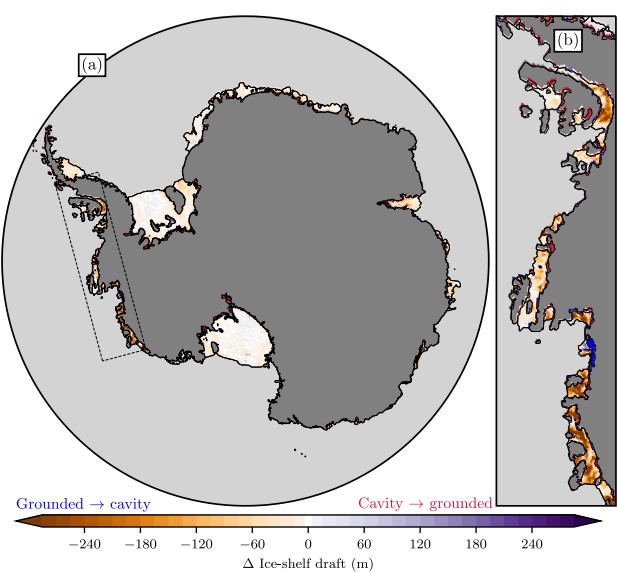

**Figure A8.** Differences between PARASO initial draft geometry taken from the f.ETISh initialization run (see Sect. 4.2) and BedMachine (Morlighem et al., 2020) (PARASO - BedMachine) for (a) the full Antarctic and (b) a Western Antarctica zoom-in. Blue nodes indicate grounded columns in the BedMachine dataset that become cavities in the PARASO initial geometry. Red nodes indicate the opposite (cavity columns turning into grounded ones).

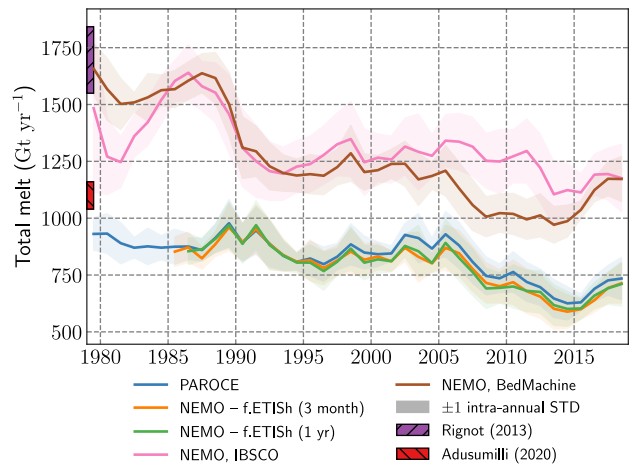

**Figure A9.** Integrated global Antarctic subshelf melt rates from different experiments (lines) and observational estimates (bars). "PAROCE" is presented in Sect. 5.2. The two "NEMO – f.ETISh" experiments share the same initial bathymetry derived from the initialization described in Sect. 4.2, and feature ocean – ice sheet coupling (no CCLM$^2$) at distinct frequencies (3 month and 1 year). "NEMO, IBSCO" and "NEMO, BedMachine" are obtained from NEMO stand-alone experiments, with bathymetry and static cavity geometry taken from the International Bathymetric Chart of the Southern Ocean (Arndt et al., 2013) and BedMachine v2 (Morlighem et al., 2020), respectively. Observational datasets are taken from Rignot et al. (2013) and the steady-state of Adusumilli et al. (2020).



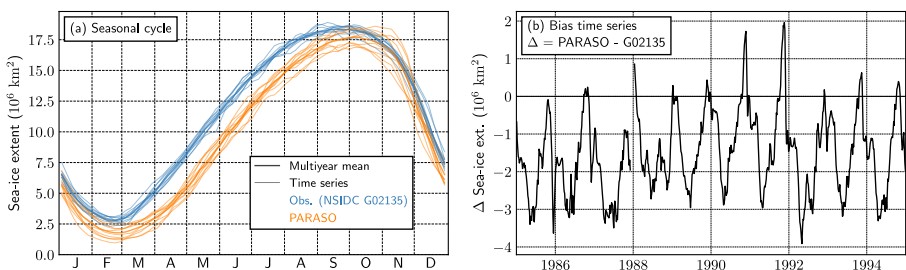

**Figure A10.** Antarctic sea-ice extents from observations (NSIDC-G02135, Fetterer et al., 2017) and from an additional fully coupled PARASO experiment (not presented here) covering the 1985 - 1994 period: (a) seasonal cycle and (b) bias time series (PARASO - NSIDC-G02135). In (b), the gap around 1988 is due to a shortage in observational data.



## Appendix B: Computational aspects

### B1  Numerical stability

While only one two-year simulation is presented in this paper, a considerable amount of sensitivity and tuning experiments have been performed for this study. In total, over 50 years of equivalent fully simulated years have been performed with the final PARASO source code, mostly for limiting the large bias developments. With these "latest" sources (distributed along this paper), no numerical instability has been observed yet, which suggests that PARASO can be considered as numerically stable. In our case, this was achieved by:

- Reducing the COSMO time step from its AEROCLOUD (Souverijns et al., 2019) value of $120$ s to $90$ s, in order to avoid CFL-criterion errors. Such errors may probably linked to the finer PARASO vertical resolution compared with AEROCLOUD ($+50\%$).

- Adapting the land "transfer coefficient limiter" implemented in COSMO by Davin et al. (2011). Over land, CLM computes all fluxes (including turbulent ones) and sends them to COSMO. However, the COSMO vertical physics scheme requires transfer coefficients ($C_D$, $C_H$) and surface properties ($T_S$, $q_s$) instead of fluxes *per se*. Hence, transfert coefficients are re-evaluated from CLM-originating fluxes through dividing by the near-surface temperature and moisture gradients (for sensible and latent heat, respectively). In cases where the surface temperatures (moistures) are very close to near-surface ones, this may lead to floating-point instabilities (small flux divided by small gradient), which can generate very large heat fluxes and subsequent crashes. In some of our tuning simulations, such crashes have been formally identified and traced from timestep-by-timestep COSMO outputs. Hence, in our simulation, $|C_H|$ is limited to $10^{-2}$.

### B2  Performance and load balancing

The PARASO experiments presented here have been performed on the Skylake nodes of the Flemish Tier-1 Breniac cluster. We asked for 224 cores running at 2.6GHz. With these resources, the wall time was about $3.5$hr for one month of simulated time. This includes all models and their I/O, as well as the offline f.ETISh coupling interfaces. Figure B1 displays the wall time required for simulating one month and the parallel efficiency of PARASO, defined as:

$$e(n_{\mathrm{proc}}) = \frac{n_{\mathrm{proc}}^{\mathrm{min}} T(n_{\mathrm{proc}}^{\mathrm{min}})}{n_{\mathrm{proc}} T(n_{\mathrm{proc}})} \tag{B1}$$

where $n_{\mathrm{proc}}$ is the number of cores used; $n_{\mathrm{proc}}^{\mathrm{min}} = 56$ is the minimal number of cores required for PARASO (constrained by random access memory requirements); and $T(n_{\mathrm{proc}})$ is the wall time required for one month of PARASO simulated time. As Fig. B1 shows, PARASO scales reasonably well up to $\approx 10^3$ cores.

MPI-based domain decomposition parallelism is implemented in NEMO, COSMO and CLM. The CPU balancing between these models, which is shown in Table B1, has been empirically tuned. As expected, COSMO is the biggest CPU consumer, amounting for about twice as much as NEMO or CLM, which use roughly equivalent resources. f.ETISh is not shown in Table B1, since it is run sequentially inbetween NEMO – CCLM$^2$ legs. Its computational cost is negligible (less than 0.1%)


**Figure B1.** (a) Wall time required for simulating one month and (b) parallel efficiency of PARASO (see Eq. B1). Note that on (a), both axes are log-scaled; on (b), the $x$-axis is.

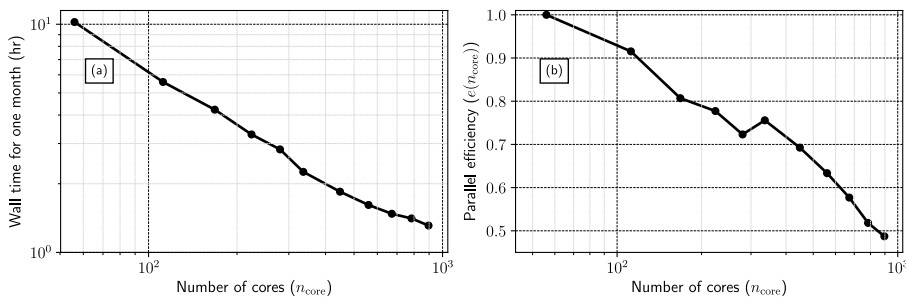

**Table B1.** Intermodel load balancing. XIOS is the NEMO I/O program, which is ran as a separate executable. f.ETISh is not listed here, as it is run inbetween NEMO-CCLM$^2$ legs.

| Model | Core | |
|---|---|---|
| | # | % |
| COSMO | 121 | 54.0 |
| CLM | 49 | 21.9 |
| NEMO | 52 | 23.2 |
| XIOS | 2 | 0.9 |

compared with all other models listed in Table B1. The same applies for the f.ETISh-including offline coupling interfaces: the equivalent walltime they require is negligible compared to what is shown in Table B1. This was to be expected, as these coupling interfaces only pre-/post-treat 2D fields at relatively low temporal frequencies (monthly at worse).

## B3    Replicability and restartability

Replicability and restartability are highly desirable properties for scientific models, but achieving them remain a considerable computer science challenge in the context of climate models (Massonnet et al., 2020). Like most climate model configurations, PARASO feature restart procedures, during which the content of each model's prognostic variables are written to disk in order to interrupt the execution and limit the wall time requirements. In PARASO, the restart procedure is also needed for the f.ETISh coupling, which is performed offline in between restarts. In other words, even with unlimited CPU time, PARASO

would require restarts.

Restart procedures have been implemented in all used models, plus OASIS, which writes the content of the coupler at the end of a leg for ensuring smooth restartability. Hence, PARASO restarts should theoretically be fully transparent (i.e., not have an impact on model trajectory), but empirically, it has been observed that this was not the case. Further experiments have identified COSMO as a potential source for non-restartability. Floating-point rounding errors (from double to single precision) employed

in restart I/O may lead to potentially diverging trajectories (Corden and Kreitzer, 2015). The magnitude of the differences in



NEMO trajectories induced by a restart is insignificant in comparison with that induced by a COSMO restart. This was to be expected, since the atmosphere is physically much more chaotic than the ocean.

While the PARASO experiment presented above has been performed on Breniac (Tier-1 cluster from the Vlaams Supercomputer Centre), the fully coupled setup has also been successfully run on Lemaitre3 (Tier-2 cluster from the French-speaking Belgian *Consortium des Équipements de Calcul Intensif*), without numerical instabilites. The non-transparency of restart procedures on one of them (Breniac) is a strong argument supporting the non-replicability across distinct architectures. It should however be noted that roughly speaking, simulations results from both machines were similar to each other. Moreover, outputs from identically launched experiments on one given machine-architecture combination are identical, down to bit precision.

## Appendix C: CCLM$^2$ differences between AEROCLOUD and PARASO

In this appendix, we list the major differences in the CCLM$^2$ model (tuning parameters, domain, parameterizations) between AEROCLOUD (a previous CCLM$^2$ stand-alone Antarctic configuration, see Souverijns et al., 2019) and PARASO.

The number of vertical levels in PARASO has been increased from 40 to 60. It allows a better representation of the stable boundary layer over the Antarctic ice sheet (Genthon et al., 2013; Handorf et al., 1999; King et al., 2006). The lowest full model level is located at $5\,\mathrm{m}$ ($10\,\mathrm{m}$) in PARASO (AEROCLOUD) and 14 PARASO levels (11 in AEROCLOUD) are located on the bottom $1000\,\mathrm{m}$ of the atmosphere. To prevent numerical instabilities related to this new vertical resolution from arising, the COSMO time step has been decreased from $120\,\mathrm{s}$ to $90\,\mathrm{s}$.

The spatial domain in AEROCLOUD is limited to the Antarctic ice sheet. To study the Antarctic climate variability at the decadal timescale, the PARASO domain has been extended to the whole Southern Ocean to explicitly simulate the interactions between the atmosphere and the ocean. The boundaries of the PARASO domain are thus located between $50°\mathrm{S}$ and $40°\mathrm{S}$. The horizontal resolution is unchanged ($0.22°$).

In order to improve the representation of perennial snow and land surface processes, some Antarctic-specific adjustments have been made in AEROCLOUD. These adjustments concern:

1. the representation of the snowpack in the Community Land Model following van Kampenhout et al. (2017);

2. the roughness length of snow to have a correct representation of the katabatic winds at the coastal margins of the Antarctic ice sheet and the ice shelves;

3. the turbulence scheme to account for the strong stable conditions.

More information can be found in Souverijns et al. (2019). The inclusion of the AEROCLOUD two-moment cloud-microphysics scheme is disregarded in PARASO, because over this extended domain and with the increased number of vertical levels, only small improvements were found in the representation of clouds and solid precipitation. The computational resources necessary are much larger than those needed for the default parameterization scheme based on the one moment Kessler scheme used to calculate the effects of grid-scale clouds and precipitation.





Compared to AEROCLOUD, the spectral nudging has been switched off. A prime objective of future work with PARASO is to study the mesoscale interactions and their potential impact on larger-scale atmospheric and oceanographic structures. Accordingly, we did not impose any relaxation in the inner domain of the atmospheric model towards the large-scale driving model (ERA5). In this way, the CCLM$^2$ dynamics is preserved.

PARASO uses the default snow roughness length of CLM4.5 of $2.4 \cdot 10^{-3}$ m, which is constant in time and space. This contrasts with AEROCLOUD, for which the snow roughness length had been set to $10^{-5}$ m. Moreover, PARASO uses the standard snow and ice module of CLM4.5 (Oleson et al., 2013) and does not include the modified scheme implemented in AEROCLOUD. Future sensitivity studies of the modified snow and ice scheme and of the value for the surface roughness length are planned.

In COSMO, the turbulence scheme uses a 1D closure with a prognostic equation for Turbulent Kinetic Energy (TKE). To better represent very stable conditions over the Antarctic ice sheet, a reduction of the limit on the vertical diffusion coefficients has been considered in both AEROCLOUD and PARASO, based on Cerenzia et al. (2014). Practically, the minimal diffusion coefficients for vertical scalar (heat) transport is set to $0.05 \ \mathrm{m^2\,s^{-1}}$ and the effective length scale of subscale surface patterns over land (used to compute the energy transfer from subgrid scale coherent eddies to turbulent scale, and referred to as "thermal circulation term" in Cerenzia et al., 2014) is set to $20$ m. Note that the values provided in Souverijns et al. (2019) for those two variables are incorrect and that in the AEROCLOUD model integrations, the values mentioned above were used.

## Appendix D: CCLM tiling

Here we describe the pseudo-averaging of ocean CCLM surface properties in order to conserve local energy in the subgrid-scale tiling. The pseudo-averaging operation has an impact on the CCLM-perceived surface temperature, surface moisture, turbulent transfer coefficients, laminar transfer coefficients, lowest-level TKE, lowest-level heat and momentum diffusivity, albedo and radiative surface temperature.

Throughout this appendix, the index $j = 0, 1$ distinguishes both tiles, with $j = 0$ corresponding to the open ocean and $j = 1$ to the sea ice. $(f_0, f_1) \in [0; \ 1]^2$ are the tile concentrations and satisfy $f_0 + f_1 = 1$.

The main novelties are linked to the TKE-derived surface transfer scheme (Raschendorfer, 2005). On every cell, this scheme updates:

- $\kappa$, the TKE on the lowest level (defined on cell vertices), also used as input;

- $\mathcal{K}_m$ and $\mathcal{K}_h$, the momentum and heat diffusivities, on the lowest level (defined on cell centers), also used as input;

- $C_d$ and $C_h$, the turbulent momentum and heat transfer coefficients (in CCLM, the moisture transfer coefficient is $C_h$ as well);

- $z_0$, the surface roughness length;

- $f_h$, the scalar laminar transfer coefficient;


– $f_v$, the laminar reduction factor for evaporation;

– near-surface diagnostics: temperature, specific water vapor content, dew-point temperature, relative humidity, winds (all
values are assessed at 2m except winds which are at 10m).

### D1     Turbulent transfer coefficients and surface properties

The pseudo-averaging operation combines surface scheme output so that the three turbulent fluxes (momentum, latent heat and sensible heat) received are the average of all tiles weighted by their respective concentrations. The pseudo-averaged heat transfer coefficient is a simple weighted average:

$$C_h^{eq} = \overline{C_h^j}^{f_j} \tag{D1}$$

where $\overline{x_j}^{f_j} = f_0 x_0 + f_1 x_1$, and $(C_h^j)$ are the tile-specific heat transfer coefficients. The pseudo-averaged surface moisture is assessed by enforcing mass-flux conservation, i.e., by determining $q_s^{eq}$ so that the mass flux absorbed by CCLM is the tile-concentration average of tile-specific mass fluxes. This yields:

$$q_s^{eq} = \frac{\widetilde{C_h^{eq}} q^{low} - \overline{F_q^j}^{f_j} \overline{T_s^j}^{f_j}}{\widetilde{C_h^{eq}} + \left(\frac{\mathcal{R}_v}{\mathcal{R}_d} - 1\right) \overline{F_q^j}^{f_j} \overline{T_s^j}^{f_j}} \tag{D2}$$

where $$\widetilde{C_h^{eq}} = \frac{C_h^{eq} \|\mathbf{u}_h^{low}\| p_s}{\mathcal{R}_d} \tag{D3}$$

$q^{low}$ is the lowest-level moisture, $(T_s^j)$ the tile-specific surface temperatures, $(F_q^j)$ the tile-specific mass fluxes linked to latent heat, $p_s$ the surface pressure, $\|\mathbf{u}_h^{low}\|$ the lowest-level horizontal wind norm, $\mathcal{R}_d$ and $\mathcal{R}_v$ the gas constants for dry air and water vapor, respectively. In case Eq. (D3) leads to $q_s^{eq} < 0$ (which in practice happens in less than once in $10^8$ calculations), we set $q_s^{eq} = 0$ and accordingly increase $C_h^{eq}$ and $\widetilde{C_h^{eq}}$ to enforce mass flux conservation.

Similarly to $q_s^{eq}$, the pseudo-averaged surface temperature $T_s^{eq}$ is assessed by enforcing sensible heat conservation, which yields:

$$T_s^{eq} = \theta^{low} - \frac{\overline{Q_h^j}^{f_j} \overline{T_s^j}^{f_j} \left(1 + \left(\frac{\mathcal{R}_v}{\mathcal{R}_d} - 1\right) q_s^{eq}\right)}{c_p^a \widetilde{C_h^{eq}}} \tag{D4}$$

where $\theta^{low}$ is the lowest-level potential temperature and $c_p^a$ the dry air heat capacity. Using Eqs. (D1) - (D4) ensures mass flux, sensible heat and latent heat quasi-conservation, with minor discrepancies being imputable to their relying on the tile-averaged
surface temperature $\overline{T_s^j}^{f_j}$ instead of $T_s^{eq}$ for evaluating a small surface pressure correction term.

The equivalent momentum transfer coefficient is:

$$C_d^{eq} = \frac{\overline{\|\boldsymbol{\tau}_j\|}^{f_j} \mathcal{R}_d \theta_{s,v}^{eq}}{\|\mathbf{u}_h^{low}\|^2 p_s} \tag{D5}$$





where $\theta_{s,v}^{eq} = T_s^{eq} \left(1 + \left(\frac{\mathcal{R}_v}{\mathcal{R}_d} - 1\right) q_s^{eq}\right)$ is the surface virtual potential temperature and $(\boldsymbol{\tau}_j)$ the tile-specific wind stresses. It should be underlined that since surface currents or ice velocities are neglected in wind stress computations, all $(\boldsymbol{\tau}_j)$ are coaligned with $\mathbf{u}_h^{low}$, hence averaging $\|\boldsymbol{\tau}_j\|$ is transparent. Equation (D5) ensures momentum flux conservation.

The pseudo-averaged surface roughness length is assessed as a geometric average, tile-concentration wise:

$$z_0^{eq} = \Pi_{j=0}^1 (z_{0,j})^{f_j} \tag{D6}$$

A geometric average is used here because $\ln z_0^{eq}$ is the quantity of interest in terms of energy transfer.

### D2 Atmosphere boundary layer

The general procedure for getting all the other surface scheme output is then to calculate them from the turbulent transfer coefficients had a non-TKE based transfer scheme been used (e.g., Louis, 1979). In case such diagnostics are not readily applicable, a simple algebraic averaging is performed. $\kappa_{bot}$, the bottom-level TKE velocity, is thus assessed from $C_d^{eq}$:

$$\kappa_{bot} = c_{tke} C_d^{eq} \|\mathbf{u}_h^{low}\| \tag{D7}$$

where $c_{tke} = \sqrt[3]{16.6}$. The bottom momentum and heat diffusivity are then updated as:

$$\mathcal{K}_m^{bot} = \max\left\{\|\mathbf{u}_h^{low}\|, u_{min}\right\} d_{bot} C_d^{eq} \tag{D8}$$

$$\mathcal{K}_h^{bot} = \max\left\{\|\mathbf{u}_h^{low}\|, u_{min}\right\} d_{bot} C_h^{eq} \tag{D9}$$

where $d_{bot} = z_0^{eq} \ln\left(1 + \frac{\delta z_{bot}}{2 z_0^{eq}}\right) \tag{D10}$

In Eqs. (D8)-(D9), $u_{min} = 10^{-2}\,\mathrm{m\,s^{-1}}$; in Eq. (D10), $\delta z_{bot} \approx 10\,\mathrm{m}$ is the lowest-level cell thickness. The heat laminar transfer coefficient is then reevaluated as:

$$f_h = \frac{1}{1 + f_{lam}} \tag{D11}$$

where $f_{lam} = \frac{\lambda_{lam}}{d_{bot}} \frac{\mathcal{K}_h^{bot}}{\nu_{h,0}} \tag{D12}$

and $\lambda_{lam} = z_0^{eq} \sqrt{\frac{\nu_{m,0}}{\mathcal{K}_m^{bot}}} \tag{D13}$

In Eqs. (D12)-(D13), $\nu_{h,0} = 2.2 \times 10^{-5}\,\mathrm{m^2\,s^{-1}}$ is the scalar conductivity and $\nu_{m,0} = 1.5 \times 10^{-5}\,\mathrm{m^2\,s^{-1}}$ is the kinematic viscosity. $f_v$, the laminar reduction factor for evaporation, is evaluated as a tile-concentration wise algebraic average.

All other near-surface diagnostics (temperature, dewpoint temperature, moisture content, winds, specific humidity), which are not used in the model's dynamics, are assessed as tile-concentration wise algebraic averages.

### D3 Radiative adjustments

Minor adjustements are also implemented in the radiative scheme transfer in order to ensure surface balance of the net shortwave and longwave radiative fluxes. The equivalent cell albedo is taken as a tile-concentration average:

$$\alpha^{eq} = f_0 \alpha_0 + f_1 \alpha_1 \tag{D14}$$





which ensures net shortwave radiation conservation. The surface temperature used in the radiative scheme is decorrelated from Eq. D4. Instead, it is evaluated so that the upward longwave radiation is the same as the tile-concentration averaged one, i.e.:

$$T_{s,rad}^{eq} = \frac{1}{\epsilon_d}\left(\epsilon_0 f_0 \left(T_S^0\right)^4 + \epsilon_1 f_1 \left(T_S^1\right)^4\right)^{1/4} \tag{D15}$$

where $\left(T_S^j\right)$ are the tile-specific surface temperatures, $\epsilon_d = 0.996$ is the CCLM default emissivity, $\epsilon_0 = 1$ is the NEMO ocean emissivity, and $\epsilon_1 = 0.95$, is the NEMO sea ice emissivity. Setting the radiative surface temperature as specified by Eq. (D15) ensures net longwave radiation conservation.

## Appendix E:  Specific PARASO tunings

Our first fully coupled simulations suffered from much larger biases than the ones presented in Sect. 5. Therefore, several sensitivity and tuning experiments (not all shown here) have been performed in order to counterbalance and reduce the coupling-
induced biases. This appendix documents the PARASO parameters that were changed from stand-alone CCLM[2] and NEMO simulations (see Table E1), and presents results from some of our tuning experiments (see Fig. E1). Here we present the following two experiments:

- – v1: no tuning, default setup;

- – v2: adjustments in: the COSMO turbulent scheme; the COSMO cloud scheme; the NEMO albedo, sea-ice rheology and
schemes.

More details on the differences between these two versions are given in Table E1. f.ETISh is not included here, since including it yielded no significant changes in terms of model bias. The results presented as the PARASO runs in Sect. 5 were obtained with v2.

One of the most significant problems that has been encountered when coupling NEMO and CCLM[2], compared to NEMO
stand-alone, is the degraded snow cover over sea ice. While the increase in surface albedo does slightly counterbalance the degraded snow cover, this aspect remains problematic in PARASO. Compared with the CORE bulk formula used in NEMO, systematic biases in surface humidity have been identified (dry bias for sea ice, moist bias for open ocean, see Fig. E2). Theoretically, the dry bias over sea ice may partly be responsible for the enhanced evaporation, yet correcting it by using the CORE humidity diagnostics only yielded slightly perceivable improvements. The COSMO heat and moisture transfer
coefficient (same coefficient) has also been slightly increased to further prevent biases related to latent heat, but once again, this only yielded slight improvements. This was to be expected, as the excess in latent heat is probably related to a moist bias in the near-surface atmosphere: hence, tuning the moisture transfer coefficients only affects the speed at which the evaporation bias develops at the ocean surface, without altering the longer-term bias from developing.

All surface albedo values were slightly increased in order to limit the ocean surface warming (and sea-ice decline) that was
observed in PARASO. While we admit that this was solely done to counterbalance the PARASO warm biases, the values used



**Table E1.** Details on PARASO tunings. For each parameter, two values or methods are given, v1 (raw, nontuned) or v2 (tuned). The "New?" column specifies whether this tuning parameter was introduced for PARASO (Y) or already preexisting in the model (N).

| Model | Comp. | Parameter | Description | Value v1 | Value v2 | Unit | New? |
|---|---|---|---|---|---|---|---|
| COSMO | Surface turb. | $Q_s^{oce}(T_s^{oce})$ | Open-ocean surface humidity | -[a] | CORE[b] | kg kg$^{-1}$ | Y |
| | | $Q_s^{ice}(T_s^{ice})$ | Sea-ice surface humidity | -[a] | CORE[b] | kg kg$^{-1}$ | Y |
| | | $C_H^{oce}$ | Open-ocean heat transfer coeff. | -[a] | $\times 1.3$[c] | - | Y |
| | Cloud | radqi_fact | Subgrid-scale variability factor for ice-cloud optical thickness | 0.5 | 0.3 | - | N[d] |
| LIM | Albedo | $\alpha_{\text{sn}}^{\text{dry}}$ | Dry snow albedo | 85 | 87 | % | N |
| | | $\alpha_{\text{sn}}^{\text{mlt}}$ | Melting snow albedo | 75 | 82 | % | N |
| | | $\alpha_{\text{ice}}^{\text{dry}}$ | Dry ice albedo | 60 | 65 | % | N |
| | | $\alpha_{\text{ice}}^{\text{mlt}}$ | Bare-puddled ice albedo | 50 | 58 | % | N |
| | Dynamics | $C_D^{\text{oce}-\text{ice}}$ | Sea-ice – ocean drag coefficient | 5 | 7.5 | $10^{-3}$ | N |
| | | $p_{\text{ice}}^*$ | Sea-ice strength param. | 2 | 4 | $10^4$ N m$^{-2}$ | N |
| NEMO | Albedo | $\alpha_{\text{oce}}$ | Open-ocean albedo | 6.6 | 8.8 | % | Y |

"Comp.": component; "Surface turb.": surface turbulence; [a]: regular COSMO scheme; [b]: surface humidity diagnostics from CORE bulk formula (Large and Yeager, 2004); [c]: regular COSMO scheme value increased by 30%; [d]: adapted from Muskatel et al. (2021).

are all within the observational range and were not overtly exaggerated. Moreover, default LIM3 sea-ice albedo parameters had mostly been tuned for Arctic simulations, suggesting that some Southern Ocean adaptations may indeed be required.

Finally, sea-ice dynamical properties were also slightly retuned to limit sea-ice compaction related to the relatively strong PARASO winds. One signature of this phenomenon is the relatively small bias in sea-ice volume (see Fig. 8(b)) compared to 865 extent: the PARASO sea-ice pack is small in area, but significant in volume, hence it is quite thick (see Fig. A7). While the increase of these sea-ice dynamical parameters are quite significant ($+50\%$), the uncertainty of their actual value are also quite high (Tsamados et al., 2014; Ungermann et al., 2017).

*Author contributions.* CP ran the numerical experiments, coordinated the PARASO developments and the writing of this manuscript. CP, SVB and SH adapted COSMO from AEROCLOUD into PARATMO. PM provided expertise on the NEMO ice-shelf cavity module and its 870 inclusion within PARASO. KH, CP and LZ developed the NEMO – f.ETISh offline coupling interface. CP developed the CCLM$^2$ – f.ETISh offline coupling interface. FK and EM provided technical support for the compilation and job submission tool used for PARASO. SH, SM, CP and LZ wrote the manuscript, with inputs from all coauthors.



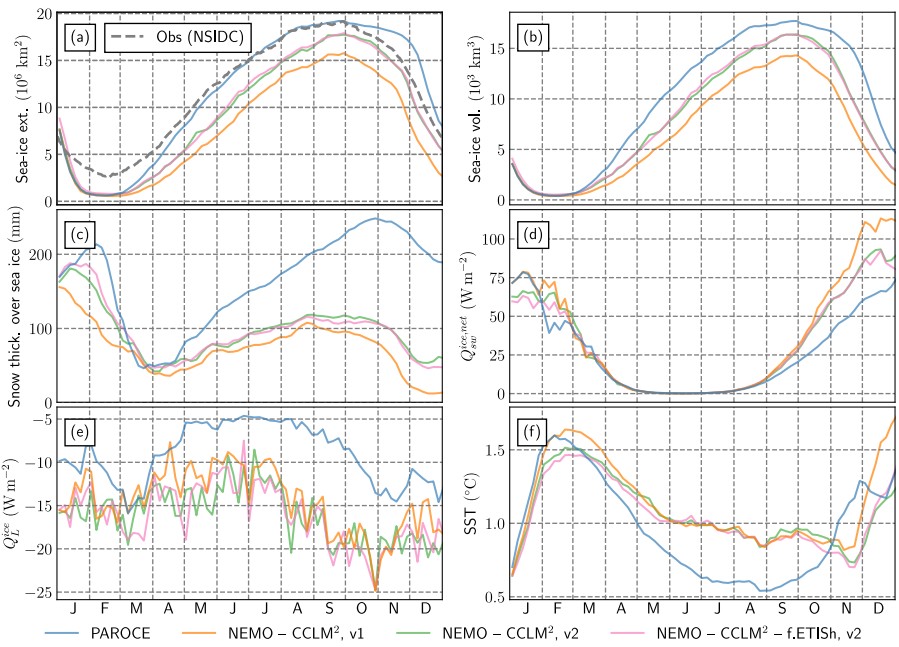

**Figure E1.** Integrated diagnostics from several PARASO tuning experiments: (a) sea-ice extent (also featuring observations from Fetterer et al., 2017); (b) sea-ice volume; (c) average snow thickness over sea ice (only sea-ice-covered cells counted); (d) net downward surface solar radiation over sea ice; (e) downward latent heat flux over sea ice; (f) average SSTs (only sea-ice-free cells counted). (c), (d) and (e) are averaged over all sea-ice covered cells. (f) is averaged over all sea-ice free cells south of $55°$S. Sea-ice coverage has been defined with a 15% concentration threshold. Results shown above are all from one-year (2000) experiments.

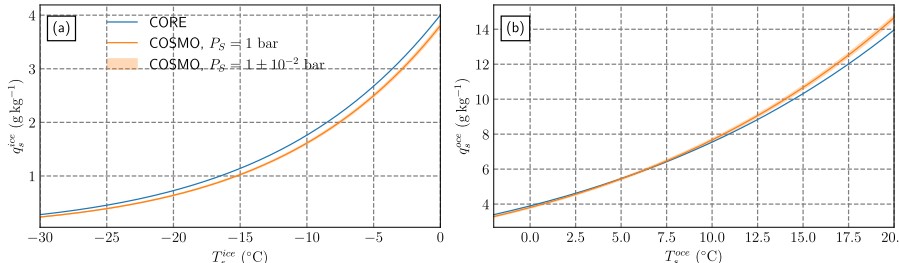

**Figure E2.** Comparison in surface moisture diagnostics (i.e., $T_s \mapsto Q_s(T_s)$) over (a) sea ice and (b) open ocean for the COSMO and the CORE (Large and Yeager, 2004) parameterizations.

*Competing interests.* The authors declare that they have no competing interests.



*Acknowledgements.* This work was supported by the PARAMOUR project, *Decadal predictability and variability of polar climate: the role*
*of atmosphere-ocean-cryosphere multiscale interactions*, supported by the Fonds de la Recherche Scientifique–FNRS under Grant number
O0100718F (EOS ID 30454083).

The computational resources and services used in this work were provided by: the VSC (Flemish Supercomputer Center), funded by the
Research Foundation - Flanders (FWO) and the Flemish Government; the supercomputing facilities of the Université catholique de Louvain
(CISM/UCL) and the Consortium des Équipements de Calcul Intensif en Fédération Wallonie Bruxelles (CÉCI) funded by the Fond de la
Recherche Scientifique de Belgique (F.R.S.-FNRS) under convention 2.5020.11 and by the Walloon Region.

Furthermore, the authors would like to thank: N. Jourdain (CNRS/IGE, France) for fostering fruitful discussions, and providing the iceberg
melt rate dataset; the JWCRP Joint Marine Modelling Programme for providing support and access to GO7 model configuration and output;
U. Blahak (German Meteorological Service) for his help with tuning COSMO; P.-Y. Barriat and F. Damien (CISM/UCL, Belgium) for their
help with Coral.
The ERA5 data (Hersbach et al., 2018) was downloaded on 01-SEP-2019 from the Copernicus Climate Change Service (C3S) Climate
Data Store. The results contain modified Copernicus Climate Change Service information 2020. Neither the European Commission nor
ECMWF is responsible for any use that may be made of the Copernicus information or data it contains.

The BedMachine Antarctica data (Morlighem et al., 2020) was downloaded on 01-FEB-2020 from the National Snow and Ice Data Center.

Most figures were created with the Matplotlib (Hunter, 2007) and Cartopy (Met Office, 2010 - 2015) Python libraries.



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
