# Peer review of "PARASO, a circum-Antarctic fully-coupled ice-sheet - ocean - sea-ice - atmosphere - land model involving f.ETISh1.7, NEMO3.6, LIM3.6, COSMO5.0 and CLM4.5"

_Geoscientific Model Development, 2021_

## Referee Comment (RC1)

Review of '*PARASO, a circum-Antarctic fully-coupled ice-sheet - ocean - sea-ice - atmosphere - land model involving f.ETISh1.7, NEMO3.6, LIM3.6, COSMO5.0 and CLM4.5*' by Pelletier *et al.*

General remarks

This paper describes a new coupled model configuration for a regional Antarctic domain. The importance of atmosphere-ocean-ice sheet feedbacks, in particular via the interactions between the ocean and ice shelf cavities, has recently spurred development of more complete coupled modelling systems and PARASO represents a significant step in this direction. This manuscript and the model it describes is well suited for publication in GMD, with the following minor corrections.

While the paper emphasises the novel nature of explicitly resolved ice shelf cavities in the coupled system, it does not do enough in the introduction to emphasise why this is so critical to achieve the scientific aims one might have when utilising such a model. A brief discussion on the impact of resolved vs. parameterised vs. absent cavities would be valuable.

Similarly, I would like the description of the chosen experiments to be placed in the main body of the text and slightly expanded upon. Experimental design should not be relegated to a table caption.

Minor corrections and typographical issues follow:

L18/19: If we are discussing WAIS ice loss, surely there is a more up-to-date reference than Shepherd et al. 2012?

L38/39: "since they allow representing the feedbacks between each component" - please rephrase.

L47: Remove the colon.

L46-50: This sentence is very long, consider breaking it up (the reference to Kruezer et al. should be a new sentence).

L51: "including ice-sheet coupling, with cavities explicitly resolved" remove comma.

L60: "is that the ice-shelf melting is" awkwardly worded. Consider changing to 'sub-shelf melt is directly computed" or similar.

L69: A reference is given for v1.0, but v1.7 is used in this study - what are the differences between these versions and do they have an impact on the results presented?

L84: While the key parameters for the ice shelves are noted in Table 1, it would be appropriate to do the same for key parameters in the ocean, or, at the very least, reference a publication where the same ocean configuration is used. The same is true for the description of the atmosphere.

L114: "we use the version 5.0" remove 'the'.

L136: "pursued' a strange choice of words, perhaps 'continued' would be more appropriate?

L158: "over 20 NEMO grid points (the full grid being ≈ 600 nodes)" Is 'grid point' and 'node' being used interchangeably here? This is not clear.

L186: "would have arisen from a category-specific coupler and the net nonsolar heat flux is a first-order development" Should this not be 'category-specific coupling' ?

L187: "While the wind stress computations do not take into account surface velocities," which velocities are being referred to here? Presumably the ocean, or the sea ice also?

L214: "Afterwards, NEMO awaits while f.ETISh" awhile should be while.

L215: Does f.ETISh also provide the heatflux of the meltwater, or is this calculated within NEMO?

L223: 'divider' should be 'divisor', I believe.

L225: should be 'ice shelf draft'. On this note, the authors are inconsistent in their hyphenating of 'ice shelf'. Please select one and make it consistent throughout.

L226: Is there a reason for the selection of a three month coupling time step? To maintain seasonality? One sentence would be nice here.

L258: Does this mean that simulating melt-induced sea level change is not possible with this configuration? I would suspect not. Do the authors consider the inclusion of dynamic sea level change to be unimportant in such a coupled system, or simply something that is not technically possible at this time?

L266: "CCLM 2 runs for one coupling time window, sending f.ETISh monthly time series of surface mass balance (SMB)". Make clear that this is offline. You say in the previous sentence that it is 'restart based' but it would be more effective if you define online and offline coupling earlier in the article, then use these consistently throughout so as not to confuse the reader with different terms that ostensibly describe the same thing.

L269: "For the sake of efficiency…" This implies a tradeoff - would there be a reason here not to do this?

L277: Section 4 feels out of place. Is there a reason the authors have chosen to describe the coupling process before the configuration of the model components? This would fit in far better when describing each individual model, in my mind.

L288: "The land ice extent is kept constant as the NEMO-COSMO interface has not been designed to deal with an evolving land-sea mask"

I think this is an important point which merits further discussion in the Discussion and Conclusions. What impact would this have for longer simulations where the grounding line and shelf retreats, but the shelf is forced to remain due to this constraint?

L351: "The CLM model starts from initial conditions that are set internally in the code Oleson et al. (2013), implying that the land component is not in equilibrium with the atmosphere at the initial time."
Remove 'implying' and rephrase - without a spinup of any kind, the atmosphere-land system is by definition not in equilibrium. "At the initial time" is also awkwardly phrased.

L354: Given that the rest of the manuscript references temperatures in degrees C, why is Kelvin used here?

L362: "keeping a maximum snow depth of 1 m is also convenient to limit the spin-up phase of the snow pack to a decade." I'm a bit confused here - above it mentions that the land is initialised from hard-coded initial conditions, but here you note that the snow pack specifically is spun-up for at least a decade? Could you clarify?

L362: "snow depth would extent the spin-up phase" extend, not extent.

L362: "but also the risk of simulating permanent snow cover (difficult to correct for thicker pack) in places" please rephrase sentence fragment in parentheses.

L364: Should be 'come' not 'comes'

L372: What is meant by a coupled field? Are these not static fields that are updated once every coupling interval? This should be rephrased for clarity.

L408: I do not like that the experiment configurations are described within the caption for Table 4. A separate paragraph should be included describing each experiment and why it was necessary to the outcomes of this paper. There is no motivation presented as to why these experiments, specifically, were chosen.

L408: "List of experiments specifically designed for this study and used in diagnostics" strangely phrased. Should be something like "and from which the presented diagnostics are derived".

L413: "coincide" is the wrong word here, I think. If the aim is to describe a similar geographical melt pattern, then rephrase. If the aim was to discuss the temporal coincidence of melt, then I'm not sure what is being described here.

Figure 6. - I question the value of panels b - e. As the patterns of melt are generally very similar, would it not be more useful to make these figures anomaly plots like panels f) and g)? I do not consider this essential and defer to the authors, but it would be my preference.

L414: "less saline Eastern Antarctic surface water in PARASO (see Figs. 6(g), 9 and 10)."

To my eye, East Antarctic coastal waters have a positive salt bias in PARASO, have I interpreted this incorrectly? Could the authors please reference which panels in Figs. 9 and 10 we should be looking at?

L415: Figures 9 and 10 are being referenced before Figures 7 and 8. Please re-order your figures accordingly.

L415 (and elsewhere): What is meant here when saying melt rates have the same 'magnitude'? Could the authors please provide values for the two experiments, or at least a value describing the discrepancy between the two experiments (i.e. one is __% larger than the other)? At the moment, it is a bit hand-wavy for my liking.

L419: "The under-estimated melt rates stem likely from increased ice shelf geometry changes" What does 'increased changes' mean physically? This is unclear to me.

L421: significantly, not significant.

L427: "display a similar behavior" remove 'a'.

L434: I disagree with the use of the term 'correlation' throughout the manuscript, as it represents a very specific mathematical relationship which is not intended here. Please rephrase. Something like 'relationship' would suffice.

L451: "East Antarctica ice sheet" - Antarctic*

L499: Is this the only source of Drake Passage transport weakening or just the only one that is clearly identifiable? What is the magnitude of this counter-current?

L502: "focus of developments in the forthcoming of the model" Poorly worded, please rephrase.

L517: "The comparison to AEROCLOUD over the Ross ice shelf, in contrast, provides similar results to ERA5." please rephrase, it is not entirely clear what is meant here.

L520: "known to be a considerable challenge in polar regions." this may be a naive comment, as I'm not an atmospheric expert, but would a reference be appropriate here?

L520: "definitely" should be "definitively".

L521: "this deficiency to one of the products mentioned above." Not clear what is meant here - what is the deficiency being referred to and what does 'products' mean in this context?

L536: "interseasonality" - change to 'interseasonal'.

L558: "fewer" should be "less"

L568: "However, our results suggest that at the short timescales investigated for this technical paper, the practical impact of this particular coupling interface is minor, and that the

main features of PARASO would be reproduced with a similar NEMO - CCLM 2 coupled configuration (i.e., excluding coupling with f.ETISh)."
This of course raises a question: with the coupling having such a small impact on short time-scales and no way to judge its impact over longer time-scales (given we look at 2 years of results here), how can we judge the soundness of this coupling approach and PARASO's utility for longer time-scale simulations where the coupling provides important feedbacks?

L572: "the biases observed in PARASO are a significant drawback," I would like the authors to highlight which biases they believe are most significant here, particularly in comparison to uncoupled models of a similar complexity/ resolution. This would be a nice synthesis of the results (while you discuss key biases below, a summary - just a list - here would be nice).

L576: "the objective was to check whether the biases were affected by the coupling interfaces themselves" I may have missed it, but is this stated previously in the manuscript? This ties back into my request for a more fleshed-out description of the chosen simulations for validation.

Fig. A10: "extents" should be 'extent'.

L703: "coupling interfaces only pre-/post-treat 2D fields at relatively low temporal frequencies (monthly at worse)." 'treat' seems an odd choice of words - should this be 'process'? Similarly, 'worse' should be 'worst' but I would suggest something more appropriate like "monthly at most frequent".

L861: "overtly" is not the right word here…

---

## Author Comment (AC2)

**Response to RC1 on manuscript gmd-2021-315**

**December 16, 2021**

We thank the referee for their careful inspection of our manuscript and insightful comments, which led to significant improvements. Nearly all comments have been addressed in the new version of the manuscript. Justifications for not incorporating two specific comments are provided below. Moreover, on top of being included in the new version, some comments calling for a more elaborate answers are also directly answered in this letter, with lines referring to the track change file which will be uploaded upon editorial permission.

> *While the paper emphasises the novel nature of explicitly resolved ice shelf cavities in the coupled system, it does not do enough in the introduction to emphasise why this is so critical to achieve the scientific aims one might have when utilising such a model. A brief discussion on the impact of resolved vs. parameterised vs. absent cavities would be valuable.*

More detail on these alternatives (no cavity, parameterized cavities), and the reasons why they were discarded for PARASO, have been added from page 3, line 70 on.

> *Similarly, I would like the description of the chosen experiments to be placed in the main body of the text and slightly expanded upon. Experimental design should not be relegated to a table caption.*

> *L408: I do not like that the experiment configurations are described within the caption for Table 4. A separate paragraph should be included describing each experiment and why it was necessary to the outcomes of this paper. There is no motivation presented as to why these experiments, specifically, were chosen.*

Agreed. More detail are provided from page 21, line 510 on, in the main text body.

> *L69: A reference is given for v1.0, but v1.7 is used in this study - what are the differences between these versions and do they have an impact on the results presented?*

More detail has been given from page 4, line 89 on. Overall, on the relatively short time scales of PARASO, the combined influence of the coupling-induced novelties (including the static calving front) are negligible. In particular, even with a free calving front, the land mask would not be significantly different, and the magnitude of the feedback on the grounded ice would be comparable to numerical noise.

> *L84: While the key parameters for the ice shelves are noted in Table 1, it would be appropriate to do the same for key parameters in the ocean, or, at the very least, reference a publication where the same ocean configuration is used. The same is true for the description of the atmosphere.*

Tables A1 and A2 have been added and referred to in the main text body.

*L215: Does f.ETISh also provide the heatflux of the meltwater, or is this calculated within NEMO?*

It is calculated within NEMO, which then extracts/injects the corresponding heat from/in the ocean at each NEMO time step (not only during coupling). Strictly speaking, this is more of an ice-shelf cavity module aspect rather than a coupled one, so this has been stressed out for clarity at page 5, line 126 in Sect. 2.2.2.

*L258: Does this mean that simulating melt-induced sea level change is not possible with this configuration? I would suspect not. Do the authors consider the inclusion of dynamic sea level change to be unimportant in such a coupled system, or simply something that is not technically possible at this time?*

Simulating melt-induced sea-level change is possible with PARASO, and would even be possible without ocean – ice sheet coupling. As soon as ice-shelf cavities and melting are included in NEMO (which is possible *without* ice-sheet coupling), the mass flux related to subshelf melt is injected into the ocean, the corresponding increase of volume gets incorporated through a divergence term, and the mean sea-surface height increases. This happens as soon as ice-shelf cavities are included, regardless of whether ocean – ice-sheet coupling is included. Hence, this aspect is technically conveyed in PARASO.

The lack of conservation discussed in the manuscript is much more minor and purely related to the ocean – ice-sheet coupling:

1. A small amount of mass and heat is brought into (resp., taken out of) the ocean after a coupling episode when NEMO opens up (resp., closes up) an ocean cell in reaction to ice-sheet geometry changes. For example, upon NEMO cell opening, the 3D ocean domain will be slightly bigger, and the new cells have internal energy which were not present prior to the cell opening, so there is a slight increment of volume and heat in the system. This is only related to the cell opening and closing, not to the ice-shelf melting in itself, whose related mass flux and latent heat had already been accounted for by NEMO, at each model time step (so not only at coupling instances).

2. At each NEMO – f.ETISh coupling time step, a divergence correction is applied for numerical stability. This is equivalent to injecting/extracting water from the grounded ice-shelf base, which yields marginal mass leak/gain (and associated internal energy).

Both these conservation caveats are really minor at the scales we are looking at. NEMO has an option for enforcing conservation (by compensating for both these caveats), but since PARASO is not a global configuration (hence, our integrated ocean mass and heat are not conserved anyway), and is meant to be run over relatively short periods, we have not activated it. Thanks for raising this point, more detail has been given at page 5, line 126 and page 13, line 303.

*L266: "CCLM 2 runs for one coupling time window, sending f.ETISh monthly time series of surface mass balance (SMB)". Make clear that this is offline. You say in the previous sentence that it is 'restart based' but it would be more effective if you define online and offline coupling earlier in the article, then use these consistently throughout so as not to confuse the reader with different terms that ostensibly describe the same thing.*

Agreed. "Online" and "offline" have been defined once and for all (done at page 7, line 170) and then systematically used, which is clearer. Thank you.

*L277: Section 4 feels out of place. Is there a reason the authors have chosen to describe the coupling process before the configuration of the model components? This would fit in far better when describing each individual model, in my mind.*

Roughly speaking, Sect. 2 describes the models (pre-existing code), Sect. 3 the coupling interfaces (new code), and Sect. 4 the configuration (input data). We feel that the manuscript flows better in that way, also because the input datasets (described in Sect. 4) are interdependent, hence it makes sense to join them in specific subsections (for geometry, initialization and forcings). In our opinion, this order makes more sense with respect to the global manuscript coherence. We also thought that it would serve the readers who are mainly interested by the input dataset we use, so that they can directly go to Sect. 4.

*L288: "The land ice extent is kept constant as the NEMO-COSMO interface has not been designed to deal with an evolving land-sea mask"*

*I think this is an important point which merits further discussion in the Discussion and Conclusions. What impact would this have for longer simulations where the grounding line and shelf retreats, but the shelf is forced to remain due to this constraint?*

This indeed is an important limitation which was not further discussed. It is one key-point that would keep PARASO from being directly transposed to longer time scales (e.g. centennal). More detail on the implications has been given at page 15, line 341. We have found that this was the right spot for it, rather than the conclusion. The latest NEMO version does allow evolving surface masks, but making it compatible with the atmosphere coupling is still a considerable challenge. The ocean – atmosphere interpolation weights (computed by OASIS in PARASO's case) would then have to be updated at each ocean – ice-sheet coupling episode.

*L362: "keeping a maximum snow depth of 1 m is also convenient to limit the spin-up phase of the snow pack to a decade." I'm a bit confused here - above it mentions that the land is initialised from hard-coded initial conditions, but here you note that the snow pack specifically is spun-up for at least a decade? Could you clarify?*

The snow pack is not spun up in the PARASO runs, but it will certainly be in future work that we plan using this configuration. We decided to drop the first year of the simulation in Sect. 5 to account for that. We expect one year to be enough, because the interactions between the atmosphere and the snowpack mainly affect the snow state in the top few meters (on short time scales; a year or so). One year is also enough, because we started our simulation from a prescribed 1 m-SWE-thick snow pack. In reality, developing such a snow pack would require 10 years where the precipitation rate is 100 mm per year (Antarctic Plateau). This approach is considered OK for a 1m SWE snowpack, because the first snow meters do show an annual cycle in temperature representative of the atmospheric forcing conditions. Thicker snowpacks would need more time to adjust to the atmospheric forcing. Accordingly, a spin-up phase of a decade should be regarded as an upper limit for a 1 m SWE snow pack to be at equilibrium. The text has been modified accordingly at page 19, line 458.

*L362: "but also the risk of simulating permanent snow cover (difficult to correct for thicker pack) in places" please rephrase sentence fragment in parentheses.*

This sentence has been removed, as it was relevant only for thicker simulated snow packs which would require a spin-up time longer than the experiments presented in the manuscript.

*Figure 6. - I question the value of panels b - e. As the patterns of melt are generally very similar, would it not be more useful to make these figures anomaly plots like panels f) and g)? I do not consider this essential and defer to the authors, but it would be my preference.*

Agreed. Figure 6 has been redesigned: for the Ronne-Filchner and WAIS, it now features PARASO absolute melts (to show that the WAIS is melting fast, and the refreezing pattern properly occuring in Ronne-Filchner) and PARASO - PAROCE anomalies (to show the relatively minor impact of coupling).

*L415: Figures 9 and 10 are being referenced before Figures 7 and 8. Please re-order your figures accordingly.*

Figures 9 and 10 refer to ocean variables, hence they are contained within Sect. 5.2. We have kept the ordering as it was, since the early reference to Figs. 9 and 10 is simply a nod supporting more elaborated comments on Fig. 6, which is in Sect. 5.1. Figures 9 and 10 are then discussed in more detail later, in Sect. 5.2. Since the GMD author guidelines do not explicitly request the figures to be numbered in order of appearance, and since it made more sense to us, we have kept the ordering as it previously was.

*L499: Is this the only source of Drake Passage transport weakening or just the only one that is clearly identifiable? What is the magnitude of this counter-current?*

It is the only one that has been clearly identified, and it also corresponds to a challenge for ocean models, and NEMO in particular (see the DRAKKAR 2021 meeting report). The magnitude has been added and commented at page 30, line 626.

As specified in the quoted sentence, this is a technical paper describing the configuration, its capacities and limitations. We believe that showing a two-year simulation, during which 8 ice-sheet coupling episodes occur, is enough for assessing the soundness of the coupling approach. A reference to the decadal time scale applications has been added in the introduction at page 3, line 61. Moreover, results from longer simulations (up to a decade) are also briefly presented in Fig. A10. Regarding the impact of the ice-sheet coupling, we have found that the ice-sheet model initialization is more determining than the presence of the coupling interface, at least at the decadal timescale. This is not a result we were happy with, but this has been further explicited out at page 33, line 698 for the sake of transparency. As already hinted at in a comment above, PARASO could probably not be used as such for longer simulations because of the constant land-sea mask constraint. Fixing this is a significant technical challenge beyond the scope of our study.

*L576: "the objective was to check whether the biases were affected by the coupling interfaces themselves" I may have missed it, but is this stated previously in the manuscript? This ties back into my request for a more fleshed-out description of the chosen simulations for validation.*

This point was added in the introduction, at page 4, line 81.

---

## Author Comment (AC3)

**Response to RC2 on manuscript gmd-2021-315**

December 16, 2021

We thank the referee for their careful inspection of our manuscript and insightful comments, which led to significant improvements. All comments but two have been addressed in the new version of the manuscript. Justifications for not incorporating two comments are provided below. Moreover, on top of being included in the new version, some comments calling for a more elaborate answers are also directly answered in this letter, with lines referring to the track change file which will be uploaded upon editorial permission.

> *Introduction: you make a good point about the need to simulate the Antarctic and Southern Ocean climate in a coupled model to capture the various complex interactions. I would recommend to also mention the typical spatial resolution which is needed to capture characteristics of the climate (e.g. global climate models usually do not provide a decent Antarctic surface mass balance). This could serve to highlight the potential of PARASO.*

Agreed. This has been added in the introduction, at page 3, line 61.

> *P6, ll. 140ff: please add some more information such as: "...provides NEMO with updated ice information about the geometry of the ice shelves. " Also it could be mentioned here already that the coupling allows ice shelves to change in thickness but not in extent.*

Slightly more detail has been added at page 7, line 176, but we have not specified that ice shelves can change in thickness but not in extent. Since grounding lines can move, technically, ice-shelf extents can change (but the total ice-sheet extent cannot, since the ice-shelf front shape has to remain constant). We estimated that this as too detailed for this brief introduction.

> *P.15, l. 342: This is a bit confusing, especially, since so far the experiment PARASO and its forcing was not yet introduced. Maybe: "...NEMO stand-alone, using consistent forcing with the subsequent coupled experiment. Specifically, this is ORAS5 forcing at its lateral boundaries and an ERA5 forcing which has been processed by the NEMO-CCLM2 coupling interface." And another question came to my mind here: is this forcing which is coming out of the coupling interface identical with the forcing which is used in the coupled experiments outside of the CCLM2 domain? From P17 and the discussion I take that this is not the case- so is the coupling interface also used outside of the CCLM2 domain for the spin-up? Maybe it would be interesting to show the difference between ERA5 derived surface forcing fluxes from the coupling interface and from the CORE bulk formula.*

> *P17, l. 375: also here: is the CORE bulk formula producing fluxes different to what the coupling interface would produce?*

Thank you for pointing that out, since this is an important point which was not clear enough in the first version of the manuscript. For the spin-up, the fluxes are not strictly speaking equivalent. ERA5 input data is used for both of them, but the COSMO interface (used for coupling) leads to fluxes distinct from the CORE bulk formula used for the spinup, even with the same input dataset (in that case, ERA5). This was already briefly hinted in the conclusion of the first submission (e.g. page 33, line 727), but probably too lightly, as the above concern testifies.

The COSMO surface scheme is very specific. It requires atmospheric inputs which are quite model-dependent (e.g. the TKE on the lowest 2 atmospheric levels), and thus not available in reference products such as ERA5. Implementing it within NEMO would have required using these inputs, which are not available over the uncoupled part of the NEMO domain. Two alternative options were then considered and subsequently discarded. First, we considered forcing NEMO with fluxes coming from the COSMO surface scheme, feeding this scheme with ERA5 input variables for classical, available fields (e.g., radiation, air-temperature, winds, etc.), and COSMO-stand-alone outputs for the "COSMO-specific" ones (TKE, etc.). This was discarded as this would have:

(a) required running a stand-alone COSMO simulation over a larger domain than PARASO's COSMO domain (to cover the full NEMO domain);

(b) led to an hybrid forcing dataset, featuring inputs from both ERA5 and COSMO-stand-alone, whose coherence would be questionable.

The second considered option was to force NEMO with offline COSMO fluxes. This would have solved (b), but not (a). Moreover, with this second option the NEMO surface properties (temperatures, sea-ice concentration, albedo) would have been completely ignored, which is not ideal either, especially over sea ice. As a result, forcing NEMO stand-alone with fluxes derived from the COSMO surface scheme was not achieved and considered beyond the scope of the paper.

Since no easy solution arose, we eventually settled with the one implemented in the paper - computing the fluxes from the same input datasets, with different bulk routines. It is a clear limitation, and this has been further stressed out in this revision, e.g. at page 17, line 384, page 18, line 441 and page 19, line 444.

> *P18ff., Results: I wonder why there is no PARCLIM experiment (a coupled atmosphere-ocean experiment)? This could provide the forcing for the PARCRYO experiments, as it seems that the model drift is to larger parts related to the atmosphere-ocean interface, which makes it difficult to compare the PARCRYO and PARASOL experiment.*

This is a good suggestion. However, while PARASO is not *that* demanding in terms of CPU requirements, running an additional nearly-fully coupled experiment within the delays of this minor review process is difficult to achieve for practical reasons (lack of computational resources). However, our guess is similar to yours: if we did run such a PARCLIM experiment and provided PARCRYO with meltrates and SMB coming from it, then the ice-sheet results would be very similar to PARASO's.

> *P19, section 5.1: this section is hard to read. I think it could be more clear and better structured if the purpose of the different experiments and comparisons would be formulated before respective paragraphs (eg.: comparison to observations, identifying the drift of ice thickness after ice shelves are allowed to evolve , illustrating the effect of SMB coupling, illustrating the effect of ocean coupling, illustrating synergy in the fully coupled system).*

More explanations on the PARCRYO experimental designs and their motivations have been provided from page 21, line 510, and Sect. 5.1 has been clarified.

> *P21, Fig. 7j: please discuss this figure a bit more. The integrated surface mass balance anomaly is almost constant in the first 2.5 months- while VAF is slightly negative- was the SMB forcing of the preceding model year of opposite sign? What is the interannual variability of SMB from ERA5 or VAF by comparison?*

More detail has been added from page 24, line 555 on. Due to the shortness of the experiments, an evaluation of the interannual variability would not be very sound and is therefore omitted.